# miR-128 regulates neuronal migration, outgrowth and intrinsic excitability via the intellectual disability gene *Phf6*

**Eleonora Franzoni[1]\*, Sam A Booker[2], Srinivas Parthasarathy[1], Frederick Rehfeld[1], Sabine Grosser[2], Swathi Srivatsa[1], Heiko R Fuchs[3], Victor Tarabykin[1], Imre Vida[2], F Gregory Wulczyn[1]\***

[1]Institute for Cell and Neurobiology, Charité-Universitätsmedizin Berlin, Berlin, Germany; [2]Institute for Integrative Neuroanatomy, Charité-Universitätsmedizin Berlin, Berlin, Germany; [3]Institute for Stem Cell Research and Regenerative Medicine, Heinrich Heine University, Düsseldorf, Germany

**Abstract** miR-128, a brain-enriched microRNA, has been implicated in the control of neurogenesis and synaptogenesis but its potential roles in intervening processes have not been addressed. We show that post-transcriptional mechanisms restrict miR-128 accumulation to post-mitotic neurons during mouse corticogenesis and in adult stem cell niches. Whereas premature miR-128 expression in progenitors for upper layer neurons leads to impaired neuronal migration and inappropriate branching, sponge-mediated inhibition results in overmigration. Within the upper layers, premature miR-128 expression reduces the complexity of dendritic arborization, associated with altered electrophysiological properties. We show that *Phf6*, a gene mutated in the cognitive disorder Börjeson-Forssman-Lehmann syndrome, is an important regulatory target for miR-128. Restoring PHF6 expression counteracts the deleterious effect of miR-128 on neuronal migration, outgrowth and intrinsic physiological properties. Our results place miR-128 upstream of PHF6 in a pathway vital for cortical lamination as well as for the development of neuronal morphology and intrinsic excitability.

**\*For correspondence:** eleonora.franzoni@charite.de (EF); gregory.wulczyn@charite.de (FGW)

**Competing interests:** The authors declare that no competing interests exist.

## Introduction

Coordinating functions for microRNAs (miRNAs) are rapidly being discovered for each of the steps required for the anatomic and functional construction of the mammalian neocortex, from stem cell proliferation and neurogenesis to neuronal outgrowth and synaptogenesis. miRNAs are short, approximately 22 nucleotide RNA molecules that primarily act as antisense regulators of gene expression. The generation of the active form of miRNAs from their initial nuclear transcripts occurs for the majority of miRNAs via two RNase-mediated processing events (reviewed in *Krol et al., 2010*; *Siomi and Siomi, 2010*). While still in the nucleus, the primary miRNA transcript (pri-miRNA) is cleaved by the concerted action of the DROSHA ribonuclease and the RNA binding protein DGCR8. DROSHA cleavage releases precursor miRNAs (pre-miRNAs) with a size range between approximately 60 and 80 nucleotides that are characterized by a stem-loop secondary structure. After nuclear export, the pre-miRNA is cleaved again to generate the active, ~22 nucleotide mature miRNA by a second protein complex containing the DICER ribonuclease. Developmental regulation of miRNA expression is known to occur at each step in this biogenesis pathway (*Krol et al., 2010*; *Siomi and Siomi, 2010*).

The global reduction in miRNA levels upon conditional deletion of *Dicer* or *Dgcr8* in neuronal progenitors is associated with early defects in proliferation and migration followed by effects on neuronal morphology including dendritic arborization, spine length, and axonal outgrowth (reviewed in

**eLife digest** The unique capabilities of the mammalian brain depend on the patterns formed by spatial arrangements and connections between millions (sometimes billions) of electrically active cells called neurons, and on the connections between these neurons. During the development of the cortex, the largest part of the brain, neurons are born in stem cell areas that lie deep inside the brain, and these newly made neurons then migrate outwards to their final positions close to the surface of the adult brain.

Franzoni et al. have examined how two molecules, a small RNA called miR-128 and a protein called PHF6, control when and how neurons migrate through the cortex and then grow to form connections with other neurons as they mature. Mutations that disrupt PHF6 can cause intellectual disabilities, and one possible reason for this is that PHF6 is needed to ensure that the neurons migrate to the correction location.

Franzoni et al. now show that miR-128 can reduce the production of PHF6 and is therefore responsible for controlling when and where PHF6 is active. Studying miR-128 in detail, they show that although an inactive precursor form of miR-128 is present in stem cells and migrating neurons, the active form of miR-128 is only found in neurons that have already reached their final position in the cortex.

Franzoni et al. used genetic methods to override the switch that controls when miR-128 becomes active. When the amount of miR-128 was artificially reduced, the neurons migrated too far. Artificially increasing the amount of miR-128 had the opposite effect: both the movement of the neurons and, later, their growth were defective. PHF6 was the key to these effects: if PHF6 levels were kept close to normal, miR-128 could no longer interfere with the movement and growth of the neurons.

Further work will be required to better understand how miR-128 is turned off and on, and how PHF6 acts to control neuronal movement and growth.

*McNeill and Van Vactor, 2012*; *Sun et al., 2013*). How individual miRNAs contribute to these phenotypes is rapidly being assessed (reviewed in *Sun et al., 2013*; *Rehfeld et al., 2015*; *Siegel et al., 2011*; *Cochella and Hobert, 2012*). Two of the best-studied miRNAs with developmental roles are miR-9 and miR-124. miR-9 acts alone or together with let-7 and miR-125 to control the timing of cell fate decisions (*Shibata et al., 2011*; *Coolen et al., 2012*; *La Torre et al., 2013*). Studies on miR-124 exemplify how a single miRNA can influence neuronal specification and function at multiple levels by regulating splicing (*Makeyev et al., 2007*), transcription complexes (*Visvanathan et al., 2007*; *Cheng et al., 2009*), and epigenetic modifiers (*Yoo et al., 2009*).

Like miR-124, the brain-enriched miR-128 is highly abundant and upregulated during embryonic mouse brain development. In another parallel to miR-124, miR-128 was first proposed to act as a developmental regulator of mRNA utilization. By inhibiting the expression of two proteins active in nonsense-mediated mRNA decay (NMD), miR-128 was shown to promote neurogenesis in a cell culture model (*Bruno et al., 2011*). Additional functions for miR-128 were then reported in behavior and memory. In a study on the acquisition and suppression of fear-evoked memory, increased expression of miR-128 correlated with, and was required for, the extinction of a learned fear response (*Lin et al., 2011*). It is presently not known if regulation of NMD mediates the effects on learning, as additional regulatory targets for miR-128 were identified in this context (*Lin et al., 2011*).

The mouse genome contains two miR-128 genes, termed miR-128-1 and miR-128-2, which are positioned within introns of two homologous genes (respectively, *R3hdm1* and *Arpp21*, also referred to as *R3hdm3*, *Rcs* or *Tarpp*). The sequence and secondary structures of the precursor miRNAs produced from the two copies of miR-128 differ, but they produce identical ~21 nt miRNAs after Dicer processing. This arrangement is evolutionarily conserved among vertebrates. Recently, the phenotypes of deletion mutants for the mouse miR-128 genes were reported (*Tan et al., 2013*). The two gene copies were shown to be unequal, with miR-128-2 responsible for approximately 80% of the miR-128 level in the adult forebrain. Deletion of miR-128-2 resulted in hyperactive motor behavior and severe epileptic seizures. Selective ablation of miR-128-2 in post-mitotic neurons in the forebrain was sufficient to cause hyperactivity and seizures that could be rescued by ectopic expression of miR-128 (*Tan et al., 2013*). The phenotype of miR-128 deletion with respect to cortical development has not been determined.

To better understand the role of miR-128 in brain development, we have examined the spatial and temporal coordinates of miR-128 expression during mouse corticogenesis and in adult stem cell niches. We present evidence that post-transcriptional regulation restricts the accumulation of miR-128 to post-migratory neurons in the embryonic cortical plate and adult stem cell zones. Premature expression of miR-128 led to deficits in the radial migration and dendritic outgrowth of upper layer cortical neurons that were associated with an increase in intrinsic excitability. In contrast, inhibition of miR-128 during migration led to a shift in final neuronal positioning toward the upper boundary of the cortical plate. We identify the X-linked syndromic intellectual disability gene *Phf6* as a significant regulatory target for miR-128. Co-expression of PHF6 suppressed both the morphological and the physiological aspects of the miR-128 gain-of-function phenotype.

## Results

### Differential regulation of miR-128 biogenesis in development

As a foundation for the functional analysis of miR-128, we began by characterizing expression of the two miR-128 genes, miR-128-1 and miR-128-2 in the mouse brain. In agreement with our previous work (*Smirnova et al., 2005*), Northern blots of RNA taken from the mouse cortex at several developmental stages show that the mature, 21 nt miR-128 RNA is upregulated between embryonic day 12.5 (E12.5) and E18.5 and remains high postnatally and in adulthood (*Figure 1A*). In this experiment, we used a high-sensitivity LNA probe complementary to the mature miRNA that should also allow detection of both miR-128 precursor RNAs (*Figure 1D*). We detected a single precursor signal present at a low level that, in contrast to the mature form, remained constant at all time points tested (*Figure 1A*).

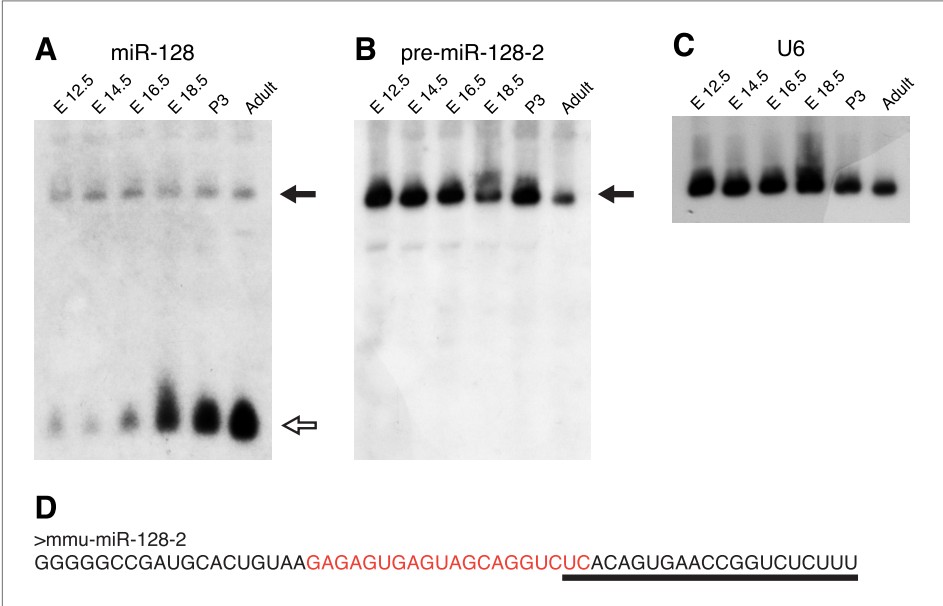

**Figure 1**. pre-miR-128-2 expression precedes miR-128. Northern blots of RNA from embryonic and adult mouse brains. RNA from the stages indicated above each lane was hybridized with probes specific for miR-128 (**A**); pre-miR-128-2 (**B**); and U6 (**C**) as loading control. The position of precursor RNAs is indicated with a filled arrow, the ~21 nt miRNA with an open arrow. The portion of the filter corresponding to ~15 to 100 nt is shown. The pre-miR-128-2 sequence is depicted in (**D**), showing the 21 nt mature sequence that is targeted by the anti-miR-128 LNA probe (underlined) and the sequence complementary to the anti-precursor hybridization probe (red).

The following figure supplements are available for figure 1:

**Figure supplement 1**. Relative activity of pre-miR-128-1-RED and pre-miR-128-2-RED expression constructs.

**Figure supplement 2**. Levels of pre-miR-128-1 are below detection level in Northern blot and in situ hybridization assays.

We next employed precursor-specific probes directed against the divergent sequences of their respective loops (*Figure 1—figure supplement 1A*). The specificity and efficacy of the two probes was confirmed using RNA from cells transfected with expression constructs for the two isoforms (*Figure 1—figure supplement 1B*). Using the pre-miR-128-2 specific probe (see *Figure 1D*), we detected a strong band of the expected size that was present at nearly constant levels throughout embryonic and postnatal development (*Figure 1B*). Expression of the miR-128-1 precursor was below the limit of detection (*Figure 1—figure supplement 2A*), indicating that miR-128-2 is more highly expressed than miR-128-1 in the embryonic cortex, consistent with a recent report (*Tan et al., 2013*). Taken together, these results suggest that the dynamic expression of miR-128 in cortical development is achieved at least in part by post-transcriptional regulation of pre-miR-128-2 processing.

## Temporal regulation of miR-128 expression during cortical development

To gain insight into the temporal and spatial dynamics of miR-128 expression, we performed in situ hybridization studies with probes specific for miR-128, pre-miR-128-1, and pre-miR-128-2 at different developmental stages. Comparing the results obtained with miR-128 and pre-miR-128-2 at E12.5, levels of miR-128 barely exceeded the detection limit (*Figure 2A*, left) despite strong precursor staining throughout the dorsal and ventral telencephalon (*Figure 2A*, middle). The pre-miR-128-1 signal, in contrast, was near or below the detection limit (see *Figure 1—figure supplement 2B*). These results are consistent with the evidence from Northern blot analysis suggesting that pre-miR-128-2 is the major expressed isoform in the neocortex and that expression of this precursor isoform precedes the accumulation of mature miR-128.

The pronounced disparity in the expression domains of miR-128 compared to pre-miR-128-2 was also apparent at later time points. In *Figure 2B*, we show representative images of in situ hybridizations performed at E14.5 with the two miR-128 probes in comparison to the neurogenic miR-124. To allow a more quantitative comparison, the average signal intensity for each probe within the combined ventricular and subventricular zones (VZ/SVZ), intermediate zone (IZ), and cortical plate (CP) was determined and expressed relative to the staining intensity of the cortical plate (*Figure 2D*). At E14.5 mature miR-128 was detected at low levels and preferentially accumulated in the cortical plate compared to the underlying subcortical zones. Staining intensity was approximately twofold (VZ/SVZ) to fourfold (IZ) lower than the CP (*Figure 2B*, left panels and *Figure 2D*, gray bars). The miR-128-2 precursor probe, in contrast, displayed an inverse pattern with almost threefold higher relative staining in the neurogenic VZ and SVZ compared to the CP (*Figure 2B*, center panels and *Figure 2D*, dark bars). Consistent with previous reports (*Cheng et al., 2009*), miR-124 was readily detected in the cortical plate but not the VZ or SVZ (*Figure 2B*, right panels). Within the IZ, an intermediate level of staining was seen (approximately 60% relative to the CP; *Figure 2D*, white bars).

These differential expression patterns were more striking at E16.5 (*Figure 2C*). The staining for mature miR-128 remained highly specific for post-mitotic neurons in the CP (*Figure 2C*, left panels) compared to the widespread presence of pre-miR-128-2 from the VZ to the CP (*Figure 2C*, middle panels). Like miR-128, miR-124 displayed uniform, high-level expression in the CP (*Figure 2C*, right panels). Unlike miR-128, however, overall levels in the IZ were intermediate compared to the lack of staining in the VZ/SVZ. Individual highly stained miR-124+ cells scattered within the SVZ and IZ may represent migrating neurons, as discussed below.

For the quantification at E16.5, the average staining intensities of the upper and lower cortical plate for the three probes were also compared (*Figure 2E*), to highlight the higher degree of deeper layer compared to upper layer staining we consistently observe using the probe for miR-128 (*Figure 2E*, gray bars). Comparing miR-128 with pre-miR-128-2, the difference in relative staining intensities in the VZ/SVZ and in the IZ was highly significant (*Figure 2E*, dark bars). Similarly, a significant difference in the relative staining of miR-124 compared with miR-128 was observed in the IZ (*Figure 2E*, white bars).

A similar difference in pattern between miR-128 and pre-miR-128-2 was also apparent at E18.5: despite uniform expression of pre-miR-128-2 throughout the cortex from the ventricles to the marginal zone, accumulation of miR-128 was restricted to the cortical plate (*Figure 2—figure supplement 1A*). In the adult, the majority of cortical projection neurons co-express the precursor and mature forms, although pre-miR-128-2+/miR-128− cells can be found scattered in the marginal zone and the subcortical white matter (*Figure 2—figure supplement 1B*).

To better characterize the subcortical cells that express pre-miR-128-2 in the absence of miR-128 during development, we repeated the hybridizations at E16.5 using fluorescent detection to allow

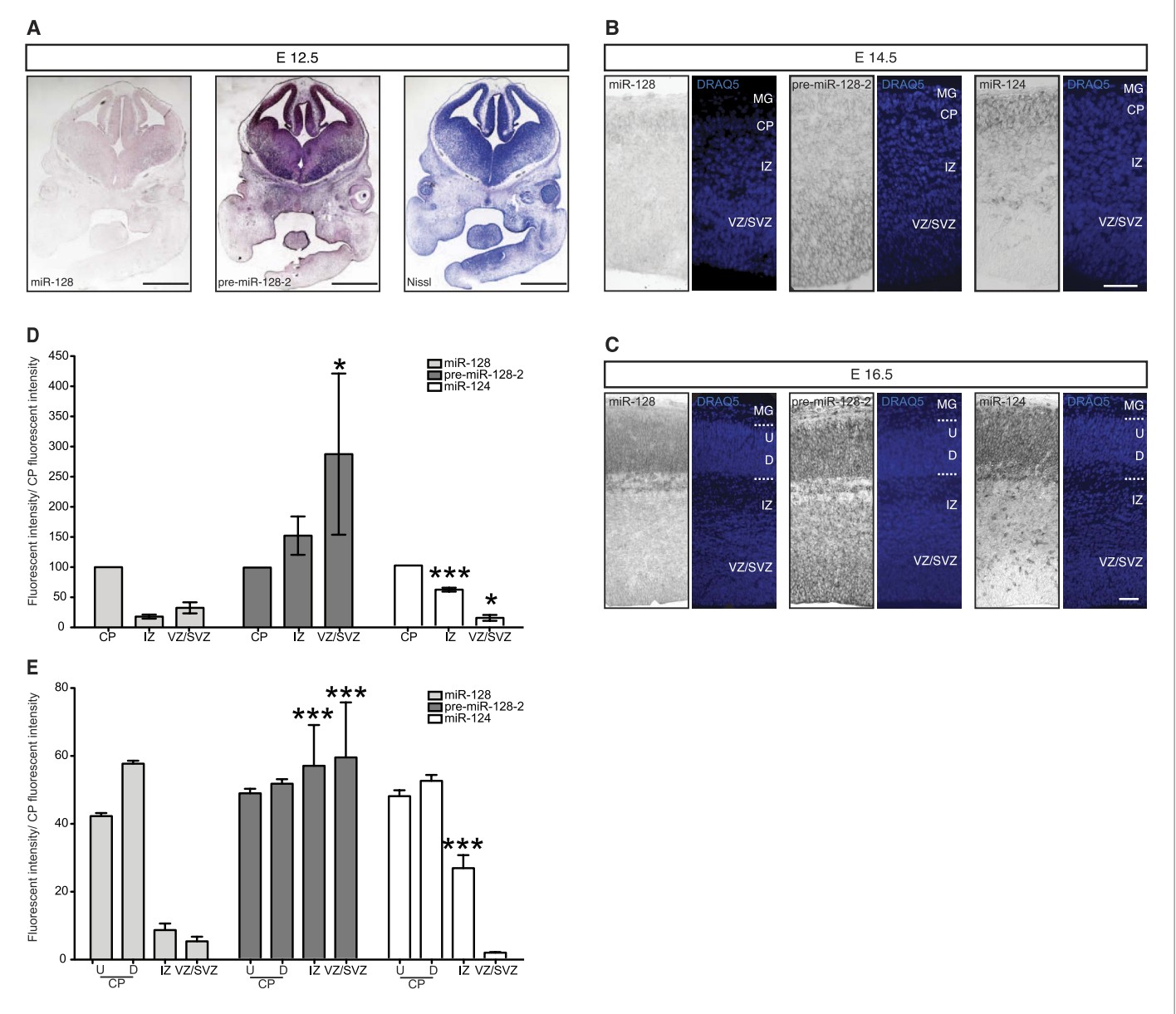

**Figure 2**. Post-transcriptional regulation determines the developmental expression pattern of miR-128. (**A**) Coronal section at E12.5 displaying embryonic telencephalon (scale bar 500 µm). Precursor staining is apparent throughout the dorsal and ventral telencephalon (middle) in the absence of miR-128 signal (left). Nissl staining is presented for comparison (right). (**B** and **C**) Coronal sections at E14.5 (**B**) and E16.5 (**C**) displaying the developing cortex stained for miR-128, pre-miR-128-2, or miR-124, as indicated. DRAQ5 staining of each section is provided for orientation. miR-128 expression is restricted to the CP at E14.5 and E16.5 (left panels) whereas pre-miR-128-2 is expressed ubiquitously from the MG to the VZ (middle panels). At E16.5 miR-128 expression within the CP shows a shallow gradient: stronger in the deep (**D**) compared to the upper layers (U). miR-124 (right) expression is detected in the CP and in some cells in the IZ. Nuclear staining is obtained with DRAQ5. Scale bar 100 µm. (**D**) Quantification of microRNA expression at E14.5 in VZ/SVZ and IZ normalized to CP (as described in 'Materials and methods'). miR-128 (gray bars) expression is highest in the CP with a reduction in the IZ (fourfold) and VZ/SVZ (twofold). pre-miR-128-2 (dark bars) expression is higher in the VZ/SVZ (almost threefold) and in IZ (1.5-fold) relative to the CP. miR-124 (white bars) is expressed in the CP and in the IZ (≈60% of the CP intensity), single positive neurons are detectable. (**E**) Quantification of micro-RNA expression at E16.5 as in (**D**) except the CP has been divided into upper (U) and deeper (D) regions using DRAQ5. miR-128 (gray bars) is expressed mainly within the CP with ninefold lower expression in the VZ/SVZ and IZ and is enriched in deeper compared to upper layer neurons in the CP. pre-miR-128-2 (dark bars) expression is higher in the VZ/SVZ and IZ (1.5-fold) compared to the CP and it is evenly distributed between upper and deeper layers. Relative distribution of miR-124 is similar to E14.5, with the IZ 10-fold higher than the VZ/SVZ. Representative false color images used for quantification are shown in *Figure 2—figure supplement 3*. Three brains per condition were analyzed. One-way ANOVA comparing miR-128 and either

*Figure 2. Continued on next page*

*Figure 2. Continued*

pre-miR-128-2 or miR-124 was performed with Bonferroni post-test. *p < 0.05, ***p < 0.001. MG: marginal zone, CP: cortical plate, IZ: intermediate zone, SVZ: subventricular zone, VZ: ventricular zone, U: upper cortical plate, D: deeper cortical plate.

The following figure supplements are available for figure 2:

**Figure supplement 1**. Differential expression of miR-128 and pre-miR-128-2 in developing and adult cortex.

**Figure supplement 2**. Post-transcriptional regulation of miR-128 during embryonic and adult neural migration.

**Figure supplement 3**. Differential staining of miR-128 and pre-miR-128-2 in corticogenesis.

antibody co-staining. Although many classical marker antibodies are not compatible with the hybridization conditions required for LNA probes (*Silahtaroglu et al., 2007*), we were able to perform co-staining for the intermediate progenitor marker Tbr2 (*Englund et al., 2005*). Within the SVZ, we found that Tbr2$^+$ progenitors stained for pre-miR-128-2 but not miR-128 (*Figure 2—figure supplement 2A,B*). As expected, Tbr2$^+$ progenitors in the SVZ did not express miR-124. Unlike miR-128, however, Tbr2$^-$/miR-124$^+$ cells could readily be detected in the IZ and SVZ, suggesting that miR-124 may be present in migrating cells (*Figure 2—figure supplement 2C*).

## Pre-miR-128-2 expression precedes miR-128 in adult stem cell niches

The absence of miR-128$^+$ cells in the embryonic subventricular and intermediate zones compared to the post-migratory neurons in the cortical plate suggests that miR-128 is not present in migrating neurons. We were interested in confirming this result in an additional developmental setting and therefore examined whether miR-128 is expressed in the migrating neuroblasts of the adult rostral migratory stream (RMS). To visualize migrating neuroblasts, we performed co-staining with Doublecortin (Dcx). The probe specific for pre-miR-128-2 strongly stained the Dcx$^+$-neuroblasts in the RMS (*Figure 2—figure supplement 2D*). In contrast, miR-128 was clearly present in the cells surrounding the RMS, but was not detectable in Dcx$^+$-neuroblasts (*Figure 2—figure supplement 2E*).

Similar results were obtained in the neurogenic niche of the adult dentate gyrus. We found that the miR-128-2 precursor was already present in newborn (Dcx$^+$/NeuN$^+$) and mature (Dcx$^-$/NeuN$^+$) granule cells of the dentate gyrus. In contrast, miR-128 was absent in immature neurons (Dcx$^+$/NeuN$^+$) and only present in mature granule cells (Dcx$^-$/NeuN$^+$) (data not shown).

In summary, we found that the miR-128-1 isoform is unlikely to contribute significantly to developmental expression of miR-128, based on the lack of signal in Northern blots or in situ hybridization (*Figure 1—figure supplement 2*). Comparing the regulation of pre-miR-128-2 and miR-128 in embryonic corticogenesis suggests that accumulation of miR-128 occurs after the completion of neurogenesis and at the end of radial migration as cortical neurons reach their final position in the cortex and begin their functional and morphological maturation (*Figure 2* and *Figure 2—figure supplement 2A–C*). Similar evidence for post-transcriptional exclusion of miR-128 from migrating neurons was obtained in the adult RMS (*Figure 2—figure supplement 2D,E*). These results prompted us to test the effects of premature miR-128 expression in embryonic progenitors as they differentiate and migrate to the cortical plate.

## Premature miR-128 expression leads to defective cortical lamination in vivo

To gain insight into the biological role of miR-128, we performed in vivo gain-of-function experiments using in utero electroporation at E15.5 to deliver ectopic miR-128 from a plasmid-based expression construct. This allowed us to introduce miR-128 into proliferating and migrating cells that normally do not express the mature miRNA. We used the plasmid vector Intron-RED, which allows precursor miRNA sequences to be expressed from a synthetic intron engineered in dsRed, generating the expression constructs pre-miR-128-1-RED and pre-miR-128-2-RED for the two miR-128 precursors (see 'Materials and methods' for details). Comparing the activity of the two constructs in Northern blot and sensor assays revealed that the pre-miR-128-1-RED construct displayed reduced activity compared to pre-miR-128-2-RED (refer to *Figure 1—figure supplement 1B,C*). The defect in pre-miR-128-1 processing therefore appears to be intrinsic to the precursor and/or flanking sequences and allows the use of the

pre-miR-128-1-RED construct as a negative control. To verify that forced miR-128 expression from pre-miR-128-2-RED can overcome the inhibitory mechanism that acts on endogenous pre-miR-128-2, we stained for mature miR-128 in electroporated brains at E18.5. We could confirm that cells expressing dsRed from the pre-miR-128-2-RED expression vector, but not the control Intron-RED vector, were the sole miR-128+ cells in the IZ (*Figure 3—figure supplement 1*).

We tested the effect of premature miR-128 expression at P7, when migration into the cortex is completed. We found that the distribution of control (Intron-RED) and pre-miR-128-1 expressing neurons was indistinguishable, with the majority of cells positioned within layers II and III (*Figure 3A*). In comparison, the majority of pre-miR-128-2 expressing neurons migrated successfully into the cortical plate but their final position was shifted toward the deep layers (*Figure 3A*). Quantification of the effect on migration confirmed the shift in neuronal position to deeper layers in response to premature expression of miR-128-2 (*Figure 3B*). These results are the first evidence that miR-128 regulates the process of radial neuronal migration during the establishment of cortical lamination.

### Premature miR-128 expression does not affect upper layer neuron specification

To gain insight into the mechanism of the migration defect, we first tested if premature miR-128 expression affects migration indirectly by interfering with the specification of upper layer neuron identity. The layer II-III neurons targeted by electroporation at E15.5 characteristically express the transcription factors Cux1 and Cux2, while earlier born layer V neurons express Ctip2 (*Nieto et al., 2004*; *Arlotta et al., 2005*). Co-staining of electroporated brains at P0 with these layer-specific markers revealed that the majority of Cux1+ cells had reached their destination in the upper layers, but some Cux1+ cells were still present in the deep layers and in the white matter. Regardless of their position in the cortical plate, cells electroporated with pre-miR-128-2-RED co-stained for Cux1 at approximately the same frequency as control cells (>80%, *Figure 3C,D,E*). Furthermore, dsRed+ cells expressing pre-miR-128-2 in layer V did not express Ctip2 at higher levels than control cells (<2%, *Figure 3C,D′,E′*) suggesting that their improper localization was not an indirect consequence of temporal misspecification. Together, these results indicate that the fate of the cells electroporated with premiR-128-2 was not affected despite the defect in neuronal migration.

### Inhibition of miR-128 leads to excessive migration of upper layer neurons

To determine the effect of blocking miR-128 expression on neuronal migration, we repeated the electroporations using a so-called sponge inhibitor. Our sponge inhibitor expresses an eGFP cassette containing an array of 16 high-affinity synthetic miR-128 binding sites within the 3′ UTR under the control of the CAGGS promoter. Upon electroporation of the anti-miR-128 sponge construct at E15.5 and analysis at P7, we observed a significant shift in neuronal position toward the top of the cortical plate in sponge compared to control neurons (*Figure 3F,G*). The inverse migration phenotypes observed in upper layer neurons upon either increasing (*Figure 3A,B*) or decreasing miR-128 (*Figure 3F,G*) activity suggests that a pathway critical for correct cortical lamination is highly sensitive to the level of miR-128.

### Premature miR-128 expression leads to aberrant morphology of migrating neurons

Based on their marker expression, manipulation of the onset of miR-128 expression did not affect the temporal identity of the resulting neurons. Careful examination of the electroporated regions, however, revealed differences in the proper bipolar morphology of pre-miR-128-2+ neurons as they migrated radially through the cortical plate (*Figure 4A,B*). Because migrating neurons change morphology quickly, we analyzed control and pre-miR-128-2 electroporations performed in the same litter to avoid differences due to small variations in mating, electroporation, or sacrifice time. In controls, the majority of the electroporated neurons were already at their correct position in layer II/III, and those still migrating presented long, radially oriented leading processes (*Figure 4A*). Neurons expressing pre-miR-128-2 were more scattered throughout the cortical plate (*Figure 4B*), with the leading processes of actively migrating cells frequently branched. To quantify this result, we reconstructed randomly selected neurons located in the deep layers and therefore still in the process of active migration. Control neurons generally had a single, unbranched leading process with occasional short filopodia (*Figure 4C*, upper row). Neurons expressing pre-miR-128-2, on the other hand, were consistently more branched and also had more filopodia (*Figure 4C*, lower row). The morphology of the neurons was quantified using the number of branches and the number of filopodia per neuron as

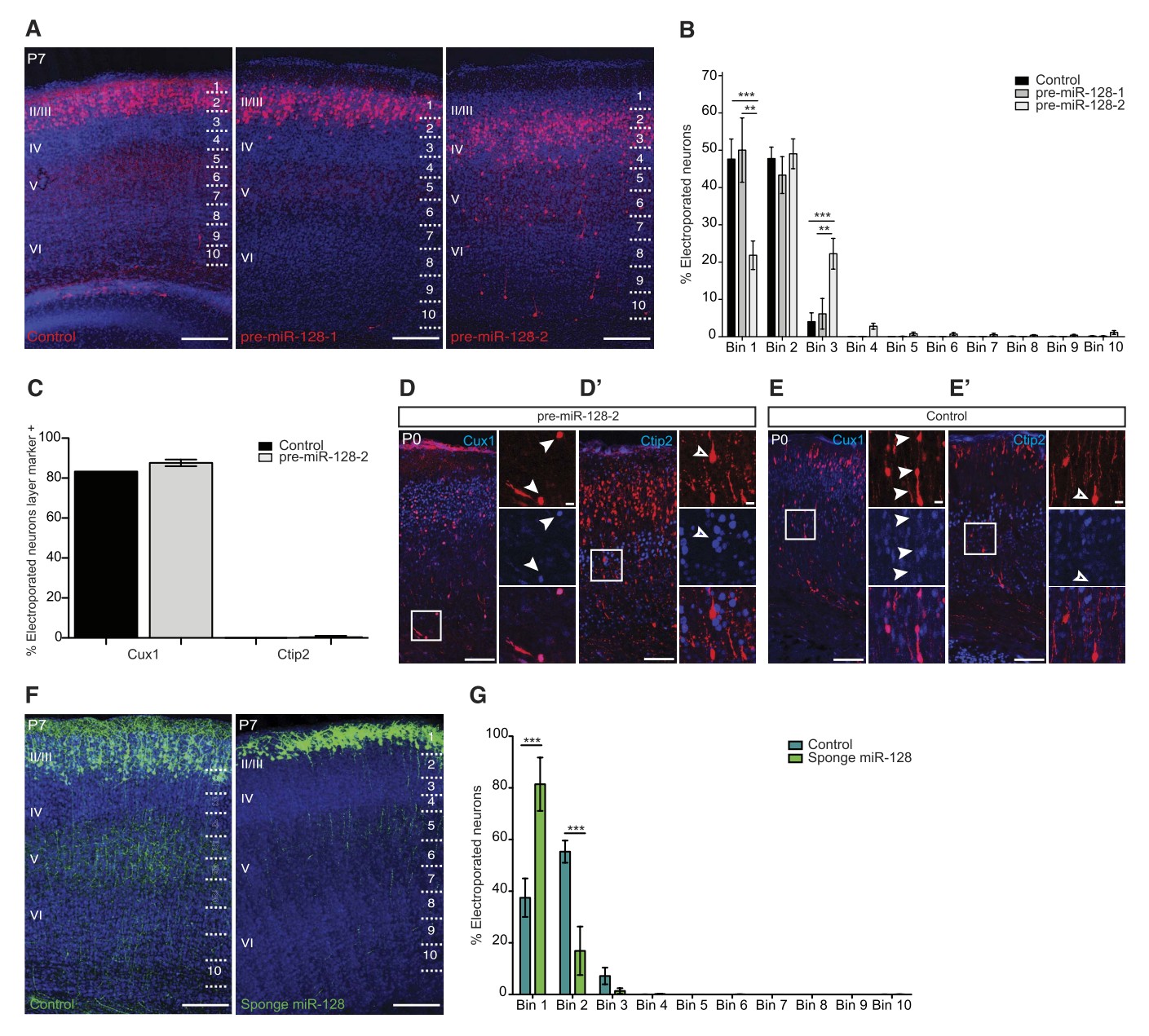

**Figure 3**. miR-128 misexpression impairs neuronal migration. (**A**) Representative brain sections of P7 mice showing intron-RED control (left), pre-miR-128-1-RED (middle), pre-mir-128-2-RED (right) after in utero electroporation at E15.5. Sections were processed for staining with DRAQ5 to reveal nuclei and anti-RFP antibody to reveal electroporated cells. On the right side of each picture the position of the bins used to assess migration is shown (see 'Materials and methods'). Scale bars represent 50 µm. (**B**) Percent of total counted neurons present in each bin is plotted. Data are from 3 to 4 mice per condition. Two-way ANOVA with Bonferroni post-test, error bars represent Standard deviation. *p < 0.05 **p < 0.01, ***p < 0.001. Electroporation of pre-miR-128-2 (white bars) but not pre-miR-128-1 (gray bars) caused a shift from uppermost layers (Bin 1) to lower layers (Bin 3) compared to control (black bars). (**C**) Quantification of P0 electroporated neurons expressing the upper layer marker Cux1 or the layer V marker Ctip2. Electroporation of pre-miR-128-2-RED (gray bars) does not change the cell fate compared to control (black bars). (**D**–**E'**) Representative brain sections of P0 mice, analyzed in (**C**), stained for dsRed to show pre-miR-128-2-RED electroporated cells (red, **D** and **D'**) and Intron-RED (red, **E** and **E'**). In (**D**) and (**E**) sections were co-stained for the layer II-IV marker Cux1 in blue. In (**D'**) and (**E'**) sections were co-stained for the layer V marker Ctip2 in blue. Neighboring images show higher magnification views of boxed regions of interest. In (**D**) and (**E**) from top to bottom: pre-miR-128-2 (red, **D**) or control (red, **E**), Cux1 (blue) and merged view. In (**D'**) and (**E'**) from top to bottom: pre-miR-128-2 (red, **D'**) or control (**E'**), Ctip2 (blue) and merged view. Scale bars 20 µm or 5 µm. Arrowheads in (**D** and **E**) mark dsRED⁺/Cux1⁺ migrating cells. Empty arrowhead in (**D'** and **E'**) marks a dsRED⁺/Ctip2⁻ cell situated in layer V. (**F**) Representative brain sections of P7 mice showing the control eGFP construct (left) and the miR-128 sponge (right) after in utero electroporation at E15.5. Sections were

*Figure 3. Continued on next page*

Figure 3. Continued

processed for staining with DRAQ5 to reveal nuclei and anti-GFP antibody to reveal electroporated cells. On the right side of each picture the position of the bins used to assess migration is shown (see 'Materials and methods'). Scale bar represents 50 μm. (**G**) Percent of total counted neurons present in each bin is plotted. Data are from 3 to 5 mice per condition. Two-way ANOVA with Bonferroni post-test, error bars represent Standard deviation ***p < 0.001. Electroporation of the miR-128 sponge caused a shift from Bins 2–3 to Bin 1 (light green bars) compared to control (dark green bars).
The following figure supplement is available for figure 3:

**Figure supplement 1**. Ectopic miR-128-2 is processed to miR-128 after in utero electroporation.

criteria (see Materials and methods for details). Consistent with their overall morphology, ectopic expression of miR-128 in migrating neurons led to an approximately 2.5-fold increase in the number of filopodia and a commensurate increase in branch number (*Figure 4D*). This suggests that the effects of miR-128 on migration are related to a failure in the regulation of cytoskeletal dynamics believed to be responsible for radial movement (*Heng et al., 2010*; *Cooper, 2013*). Staining of the electroporated area with the radial glia marker Nestin did not revel any obvious changes in the glial scaffold directing migration of these neurons, consistent with a cell autonomous effect (data not shown).

## Identification of the Börjeson-Forssmann-Lehmann Syndrome gene *Phf6* as a regulatory target for miR-128

To identify regulatory partners for miR-128 that might be responsible for the altered migration, we used prediction algorithms (TargetScan, Pictar, Diana-microT) to screen for potential target genes with known or suspected roles in neuronal migration or outgrowth (*Krek et al., 2005*; *Friedman et al., 2008*; *Maragkakis et al., 2009*). A reporter assay was used to validate sensitivity to miR-128 for the candidate genes *Gria3*, *Jip3*, *Nrp2*, *Pard6b*, *Phf6*, *Reelin*, and *Srgap2* (*Figure 5—figure supplement 1*). Of these candidates, *Pard6b* and *Phf6* were also >0.5-fold downregulated in a microarray screen of mRNAs affected by miR-128 overexpression in P19 embryocarcinoma cells (data not shown). We concentrated on the Börjeson-Forssmann-Lehmann Syndrome gene *Phf6* based on its expression pattern in the

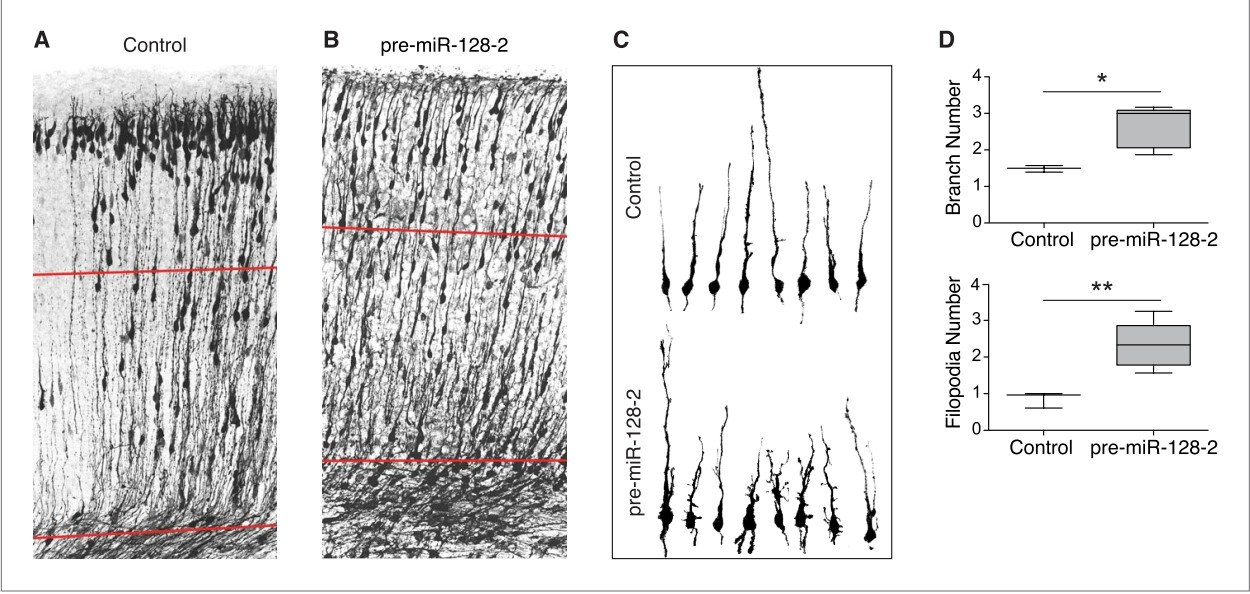

**Figure 4**. Neurons misexpressing miR-128 show impaired radial morphology. (**A** and **B**) P0 sections from littermates electroporated at E15.5 with control Intron-RED (**A**) or pre-miR-128-2-RED (**B**) expression constructs. Sections were stained for dsRed to reveal electroporated cells, rendered in black and white. Red lines indicate the boundaries of the deep layers of the cortical plate, as determined by nuclear staining (not depicted). (**C**) Reconstructed migrating neurons sampled from the deep layers (red lines in **A** and **B**). Upper row shows Intron-RED control neurons, bottom row shows pre-miR-128-2-RED electroporated neurons. (**D**) Box plot of total branch (upper graph) and filopodia (lower graph) number per reconstructed neuron. (58 neurons from 3 Intron-RED brains and 67 neurons from 5 pre-miR-128-2-RED brains were analyzed, significance determined with an unpaired Student's t test *p < 0.05, **p < 0.01).

embryonic VZ and SVZ (*Voss et al., 2007*; *Zhang et al., 2013*) and the high degree of similarity between the reported *Phf6* migration phenotype to our results with miR-128 (*Zhang et al., 2013*).

To allow a direct comparison to the miR-128 expression pattern, we performed in situ hybridizations at E14.5 and E16.5 for *Phf6* mRNA and antibody staining for PHF6 protein (*Figure 5*). Using a *Phf6*-specific LNA probe, at E14.5 *Phf6* mRNA was detected throughout the cortex with particularly prominent expression in the intermediate zone (*Figure 5A*). At E16.5 *Phf6* mRNA was also detected in the intermediate zone, but at a reduced level relative to the cortical plate, ventricular, and subventricular zones (*Figure 5B*). Antibody staining was consistent with the mRNA expression patterns at both time points, and confirmed the presence of PHF6 protein in the IZ at E16.5 (*Figure 5C,D*). Taken together, these results indicate that PHF6 is expressed throughout the miR-128 negative regions of the VZ, SVZ, and IZ at E14.5 and E16.5 (*Figure 2B,C*; *Figure 2—figure supplement 3*), and suggest that developmental regulation of PHF6 by miR-128 may occur in the domain of co-expression in the cortical plate.

The *Phf6* mRNA contains three potential, conserved binding sites for miR-128 (*Figure 5—figure supplement 2*). The sensitivity of the mouse 3'UTR to miR-128 was confirmed in a reporter assay upon co-expression of miR-128, with the response to pre-miR-128-2-RED greater than pre-miR-128-1-RED, as expected (*Figure 5E*). To determine if miR-128 can regulate endogenous *Phf6*, we used two cell lines, HeLa and HEK-293, that express *Phf6* but not miR-128. In HeLa cells transfection with synthetic miR-128 led to a strong reduction in endogenous PHF6 protein. Transfection of two non-targeting miRNAs, let-7b or miR-125, had no effect (*Figure 5F*). Similar results were obtained in HEK-293 cells. As controls, we transfected with synthetic miRNAs for either let-7b, a non-targeting miRNA, or miR-124, a microRNA with one conserved binding site in the PHF6 3'UTR. Whereas let-7b had no effect, transfection with synthetic miR-128 consistently reduced PHF6 protein levels by an average of approximately 50% (*Figure 5G*, quantified in *Figure 5H*). Unlike miR-128, the reduction in PHF6 in response to miR-124 was not statistically significant, suggesting that the three predicted binding sites for PHF6 act cooperatively to mediate stronger repression than the single site present for miR-124.

To complement the in situ data, we used qRT-PCR to show that *Phf6* mRNA levels show a reciprocal temporal relationship to miR-128, with levels highest in the embryonic cortex and an approximately 50% reduction between E16.5 and P3 (*Figure 5I*, compare to the miR-128 profile in *Figure 1A*). A similar inverse relationship was observed during maturation of cultured embryonic cortical neurons. *Phf6* mRNA was maximally expressed in the first two days of culture and declined with increasing time in culture (*Figure 5K*). Levels of miR-128 determined in parallel showed an inverse profile with levels increasing over time in culture (*Figure 5L*). Western blots confirmed the reduction in PHF6 expression at the protein level (*Figure 5J*).

## Co-expression of the Börjeson-Forssmann-Lehmann Syndrome gene Phf6 rescues the migration defect caused by pre-miR-128-2

Zhang et al. have shown that shRNA-mediated knockdown of *Phf6* in the developing cortex led to a similar effect on radial neuronal migration and morphology as premature miR-128 expression (*Zhang et al., 2013*). To test if miR-128 might be acting via suppression of *Phf6*, we generated an expression plasmid containing the open reading frame of *Phf6* linked to eGFP via an IRES sequence. As a negative control, we tested a similar construct containing the ORF of *Nrp2*, a known regulator of migration but weak miR-128 target (see *Figure 5—figure supplement 1*). Co-expression of NRP2 and pre-miR-128-2 after electroporation at E15.5 had no effect on the migration of cortical neurons assayed at P7 compared to expression of pre-miR-128-2 alone (data not shown). In contrast, co-expression of PHF6 and pre-miR-128-2 significantly reduced the number of ectopic neurons in the lower layers and promoted their migration into the upper layers (*Figure 6A–C*). Quantification of neuronal position at P7 confirmed that significantly more PHF6/miR-128 double-positive neurons reached the upper layers than those expressing miR-128 alone (*Figure 6C*). These results suggest that precise timing of miR-128 expression is required to fine-tune the pro-migratory function of PHF6.

## Premature miR-128 expression reduces dendritic arbor complexity in upper layer neurons

The results presented so far indicate that correct temporal control of miR-128 expression is necessary to avoid interference with PHF6-mediated neuronal migration. We therefore wondered if this balance is also important for the maturation of neurons in the cortical plate. For these experiments, electroporations were performed using the same conditions as in the migration experiments but analyzed at

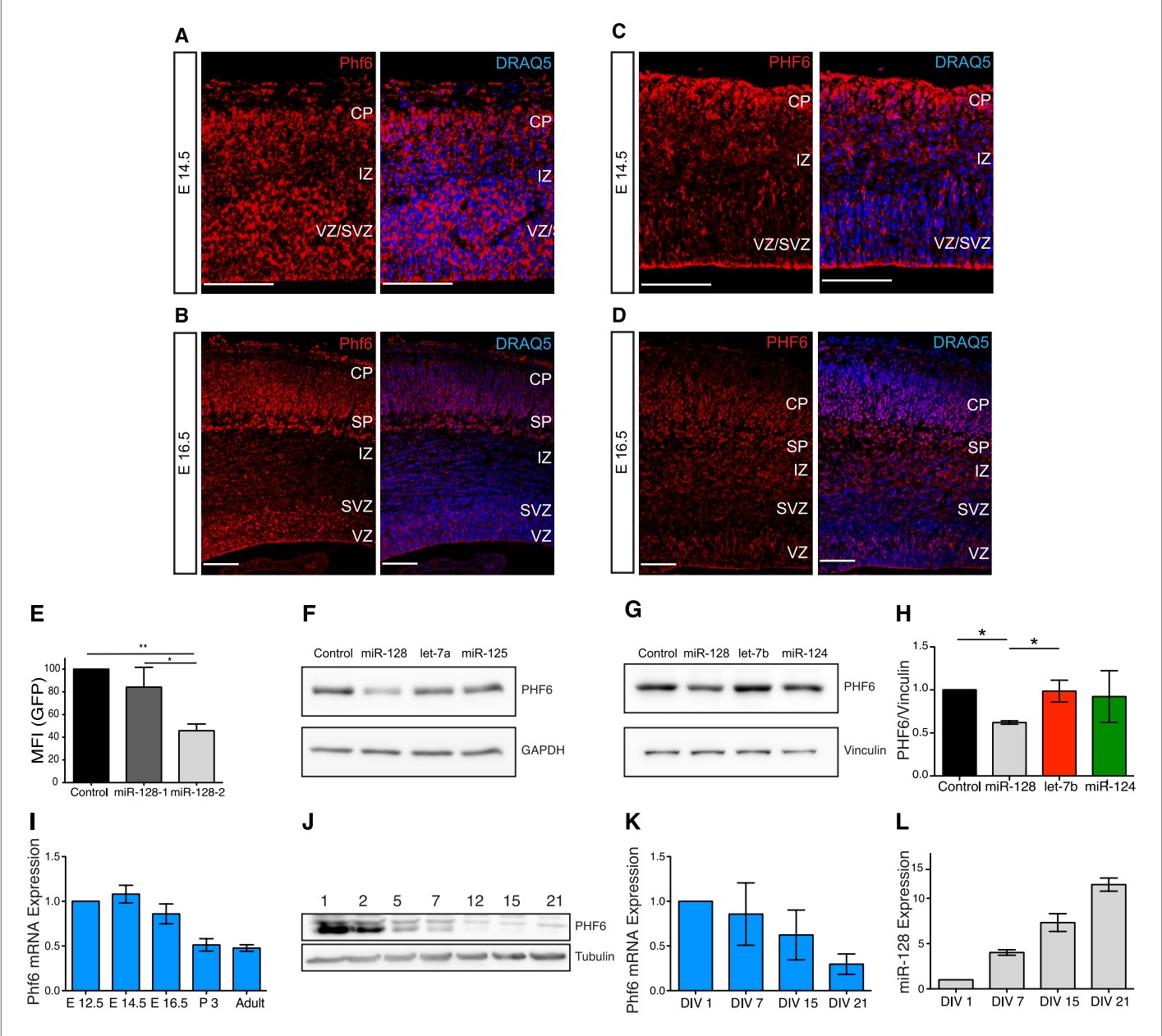

**Figure 5**. Regulation of PHF6 by miR-128. (**A** and **C**) Phf6 mRNA (**A**) and protein (**C**) expression domains in E14.5 brain are comparable, both the mRNA and the protein are present in the VZ, SVZ, and IZ. The nuclear marker DRAQ5 allows the visualization of the brain subregions. Antibody specificity is documented in *Figure 5—figure supplement 3A*. (**B** and **D**) Phf6 mRNA (**B**) and protein (**D**) in E16.5 brain section are found in the CP, IZ as well as the SVZ and VZ. mRNA and protein expression patterns are comparable. The nuclear marker DRAQ5 allows the visualization of brain subregions. Scale bar 50 μm. CP: cortical plate, SP: subplate, IZ: intermediate zone, SVZ: subventricular zone, VZ ventricular zone. (**E**) Reporter assay on the Phf6 3'UTR, cloned in an eGFP plasmid. pre-miR-128-RED expression constructs and Intron-RED control were co-transfected with the GFP-Phf6-3'UTR sensor plasmid in HEK-293 cells. The GFP Mean Fluorescent Intensity (MFI) of miR-128/Phf6-3'UTR expressing cells is normalized to the GFP MFI of control/Phf6-3'UTR expressing cells. One-Way ANOVA with Bonferroni post-test, error bars represent Standard deviation *$p < 0.01$, **$p < 0.05$. (**F**) Representative Western blot of extracts from HeLa cells transfected with scrambled control, miR-128, let-7b or miR-125 synthetic miRNA mimics, as indicated. miR-128 has three, miR- let-7b and miR-125 no predicted binding sites in the Phf6 3'UTR. Upper panel shows signal for endogenous PHF6 protein, lower panel GAPDH as loading control. (**G**) Representative Western blot of extracts from HEK-293 cells transfected with scrambled control, miR-128, let-7b or miR-124 synthetic miRNA mimics, as indicated. miR-128 has three, miR-124 one and let-7b no predicted binding sites in the Phf6 3'UTR. Upper panel shows signal for endogenous PHF6 protein, lower panel Vinculin as loading control as indicated to the right. (**H**) Quantification of PHF6 protein levels relative to Vinculin, as shown in (**F**). miR-128 expression reduced PHF6 protein levels approximately 50% compared to the let-7b control (average of 3 independent experiments, *$p < 0.01$ One-Way ANOVA, error bars represent Standard deviation). (**I**) qRT-PCR for Phf6 mRNA from staged mRNA samples between E12.5

*Figure 5. Continued on next page*

*Figure 5. Continued*

and Adult. Phf6 expression was normalized to the reference mRNA Oaz1. Average of three independent experiments, error bars show Standard deviation. (**J**) Western blot of PHF6 protein levels in primary cortical neurons cultured for the indicated days in vitro (DIV). (**K**) qRT-PCR for *Phf6* mRNA performed on primary cortical neurons, DIV as indicated. *Phf6* expression was normalized to the reference mRNA GAPDH. (Average of three independent experiments, error bars represent Standard deviation). (**L**) TaqMan qPCR for miR-128 was performed on the same RNA samples as in Panel **J**. Expression level was normalized to sno135 RNA (Average of three independent experiments, error bars represent Standard deviation).

The following figure supplements are available for figure 5:

**Figure supplement 1**. Validation of miR-128 targets using a reporter assay.

**Figure supplement 2**. Multiple, conserved binding sites for miR-128 in the Phf6 3'UTR.

**Figure supplement 3**. Western blot detection of Phf6.

---

P15. We performed whole-cell patch-clamp recordings in combination with intracellular biocytin labeling of pyramidal cells located in layer II/III and compared control (Intron-RED), miR-128 gain-of-function (pre-miR-128-2-RED) and PHF6 rescue (pre-miR-128-2-RED plus PHF6-GFP expression vectors) conditions (*Figure 7A*).

To determine the effect of miR-128 on neuronal morphology, individual dsRed$^+$ upper layer neurons were reconstructed after staining for biocytin (*Figure 7A* and *Figure 7—figure supplement 1*) and their dendritic complexity compared using Sholl analysis (*Figure 7B*). We observed a significant reduction in the number of dendritic intersections, a measure of dendritic complexity, in cells electroporated with pre-miR-128-2-RED compared to the IntronRED control. The number of dendritic intersections was reduced approximately 37% for the proximal arbors 40–75 µm from the soma (*Figure 7B*). Co-electroporation of pre-miR-128-2-RED and PHF6-GFP largely counteracted this effect of miR-128. Compared to cells electroporated with pre-miR-128-2-RED alone, a statistically significant increase in intersection numbers was observed between 40 and 75 µm from the cell body. There was no significant difference in this parameter at any distance from the soma between control cells and cells co-expressing miR-128-2 and PHF6 (*Figure 7B*).

To confirm these results, we also performed Sholl analysis on layer II/III neurons at P21, comparing control (Intron-RED) and miR-128 gain-of-function (pre-miR-128-2-RED) conditions. In these experiments individual neurons were reconstructed after staining for dsRed to amplify the fluorescent signal (*Figure 7—figure supplement 2A–C*). Cells prematurely expressing miR-128 displayed a statistically significant decrease in proximal dendritic complexity throughout the area approximately 35–120 µm from the cell body compared to control (*Figure 7—figure supplement 2D*). Neither the length nor the orientation of the apical dendrites was noticeably affected. As an additional control, we also tested the less active pre-miR-128-1-RED expression construct. As expected, Sholl analysis of the resulting neurons yielded an intermediate phenotype that was not statistically different than control (*Figure 7—figure supplement 2B,D*). This result indicates that the reduction in dendritic complexity associated with premature miR-128 expression persists after P15 and is therefore more likely due to interference with, as opposed to a delay in, dendritic outgrowth.

In addition to morphological changes, whole-cell patch clamp recordings revealed differences in the intrinsic physiological properties of layer II/III pyramidal cells in response to miR-128 gain-of-function. After electroporation of pre-miR-128-2-RED, the affected neurons had a significantly more depolarized resting membrane potential ($V_M$) than cells electroporated with the Intron-RED control ($V_M = -64.6 \pm 1.3$ mV vs $-73.0 \pm 1.4$ mV, *Figure 7C*). Furthermore, neurons prematurely expressing miR-128 showed a steeper current–voltage relationship across a range of hyper- and depolarizing current pulses compared to control cells, an effect that can be primarily accounted for by their higher input resistance ($203 \pm 18$ MΩ for pre-miR-128-2-RED vs $161 \pm 18$ MΩ for Intron-RED, 26% change, *Figure 7D,E*). Because we found no difference in the membrane time constant between pre-miR-128-2 expressing cells and control cells (data not shown), the increased input resistance is most likely a consequence of the observed reduction in dendritic complexity (*Figure 7B*). However, in combination with the depolarized membrane potential it may also indicate a reduction in basal membrane conductance

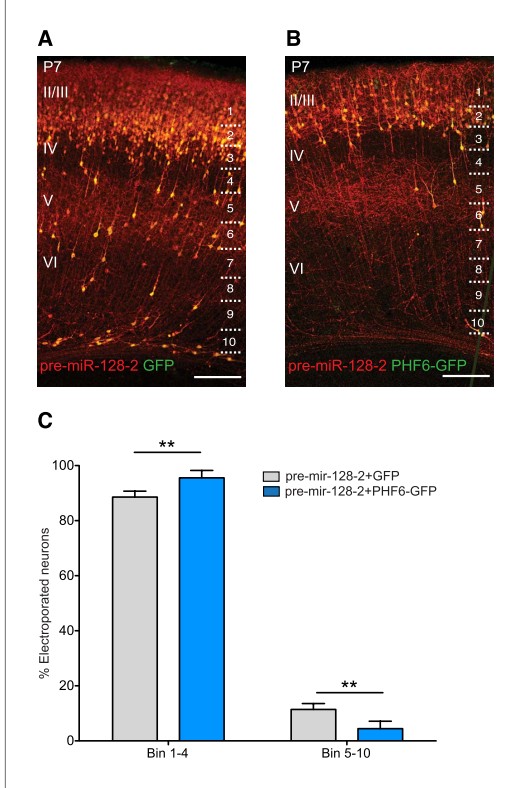

**Figure 6**. PHF6 rescues the migration defect caused by pre-miR-128-2. (**A** and **B**) Brain sections of P7 mice electroporated at E15.5 with pre-miR-128-2-RED (**A**) or pre-miR-128-2-RED plus PHF6-GFP expression constructs (**B**). Sections were stained for dsRed and GFP to reveal electroporated cells. The position of bins used to quantify migration is shown on the right. Scale bar represents 50 µm. Cortical layers are labeled on the left, as determined by nuclear staining (not depicted). (**C**) Number of neurons in each bin was determined and expressed as the per cent contained in upper layers (Bin 1–4) vs deeper layers (Bin 5–10). (Five mice analyzed per condition. Significance determined by Two-way ANOVA with Bonferroni post-test **$p < 0.01$, error bars represent the Standard deviation).

mediated by potassium leak channels. In either case, in their sum, these changes should lead to an increase in excitability. Indeed, we observed a reduction in the current required to trigger action potential discharge (rheobase) in pre-miR-128-2 expressing cells compared to control cells (79.3 ± 1.0 pA vs 124.5 ± 12.8 pA, *Figure 7F*). Furthermore, pre-miR-128-2 expressing cells fired trains of action potentials (APs) at substantially higher frequencies (42 ± 4 Hz) in response to large depolarizing current pulses (250 pA, 500 ms), a 63% increase compared to control cells (26 ± 1 Hz, *Figure 7G*). Interestingly, in response to large hyperpolarizing pulses the miR-128 gain-of-function neurons responded with an approximately twofold larger voltage sag than control neurons (5.6 ± 0.5 mV vs 2.6 ± 0.2 mV, measured for −250 pA current pulses, *Figure 7H*). This suggests that during hyperpolarization an increase in HCN-mediated $I_h$ currents may partially compensate the higher input resistance seen in the miR-128 gain-of-function neurons.

Using cells obtained from co-electroporations of pre-miR-128-2-RED and PHF6-GFP, we found that the effects of premature miR-128 expression on the electrophysiological properties of layer II/III neurons are for the most part mediated by PHF6. Neurons co-expressing PHF6 and pre-miR-128-2 had a $V_M$ of −70.5 ± 1.4 mV and an input resistance of 180 ± 16 MΩ, both comparable to that of control Intron-RED cells (*Figure 7C,E*). The current–voltage relationship, rheobase and the maximum AP discharge were also partially rescued by PHF6 co-expression (35 ± 2 Hz at 250 pA, 500 ms, *Figure 7D,F,G*). The increase in the hyperpolarization-induced voltage sag was also partially reversed by PHF6 (4.3 ± 0.8 mV), although it remained higher than in control neurons (*Figure 7H*).

In summary, miR-128 misexpression during corticogenesis results in substantive changes in both the morphological and physiological properties of upper layer neurons. With the exception of the voltage sag and rheobase, which were partially compensated, the observed reductions in dendritic complexity and changes in intrinsic excitability were restored to control levels by co-transfection with PHF6.

## Discussion

By carefully analyzing the expression pattern of miR-128 during cortical development, we present evidence that miR-128 might be part of a regulatory switch required for the transition from migration to outgrowth, thereby promoting functional neuronal maturation. Based on the disparate temporal control of pre-miR-128-2 and miR-128, post-transcriptional mechanisms appear to contribute to the timing of miR-128 activity. Post-transcriptional regulation of miRNA biogenesis is believed to facilitate dynamic control over miRNA activity that may be required for cells to rapidly change their gene expression in response to developmental or environmental signals (*Krol et al., 2010*). Another possible advantage of post-transcriptional control is that it would allow the timing of miR-128 expression to be partly uncoupled from the regulation of *Arpp21* transcription, the host mRNA for miR-128-2. One example of this in the nervous system is the ability of miR-26 to suppress its host gene *Ctdsp2*

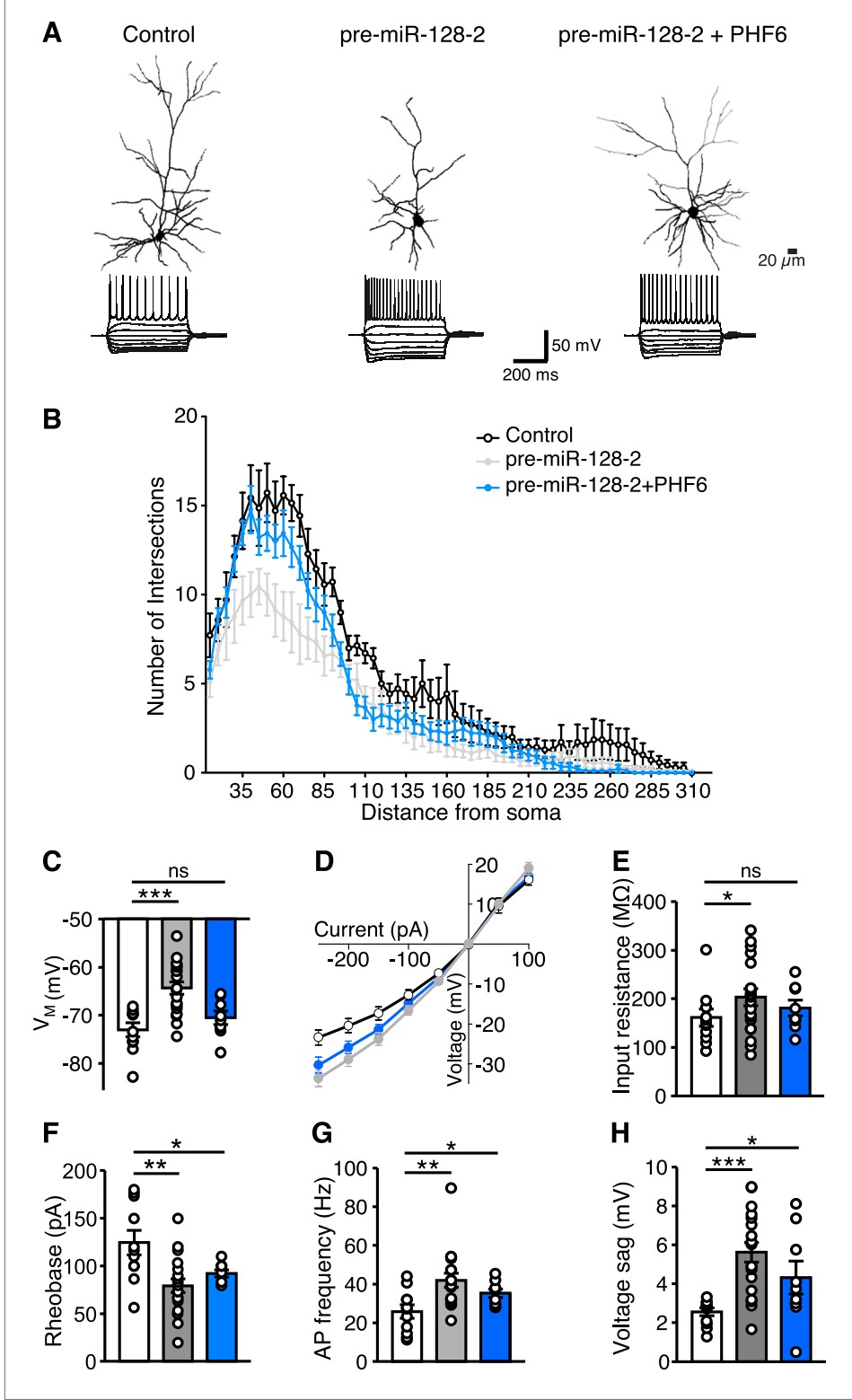

**Figure 7**. miR-128 and PHF6 regulate dendritic complexity and intrinsic excitability. (**A**) Cells from electroporations using Intron-RED (left), pre-miR-128-2-RED (middle), or pre-miR-128-2-RED plus PHF6-GFP (right) were recorded and filled. Representative reconstructed neurons (top) and their voltage responses to a family of current pulses (bottom) are shown. Compared to Intron-RED control, AP discharge is increased by pre-miR-128-2-RED and

*Figure 7. Continued on next page*

*Figure 7. Continued*

intermediate upon co-expression of pre-miR-128-2-RED and PHF6-GFP. (**B**) Sholl analysis of filled and reconstructed neurons, from Intron-RED (open circles, n = 7 cells), pre-miR-128-2-RED (gray, n = 9 cells), and pre-miR-128-2-RED plus PHF6-GFP (blue, n = 9 cells) electroporated neurons. Error bars represent standard error of the mean. (**C**, **E**–**H**) Summary bar charts of intrinsic physiological properties: Membrane potential (VM **C**), Input resistance (RI **E**), Rheobase (**F**), Action Potential (AP) frequency (**G**) and voltage sag (**H**). Colors as in (**B**), bars are overlain by data from individual cells. (**D**) Current–voltage relationship for the three groups of electroporated neurons, color scheme as in (**B**). Note the steep curve for pre-miR-128-2-RED neurons, and partially recovered RI relationship for PHF6 rescue neurons. Statistics: ns – p > 0.05, *p < 0.05, **p < 0.01, ***p < 0.001, Two-way ANOVA for graph in (**B**) and Mann–Whitney non-parametric test for graphs in **C**, **E**–**H**.

The following figure supplements are available for figure 7:

**Figure supplement 1**. Reconstructed neurons used to perform Sholl analysis at P15.

**Figure supplement 2**. pre-miR-128-2 but not pre-miR-128-1 affects dendritic arbor complexity.

and allow differentiation of neural stem cells (*Dill et al., 2012*). There is evidence for a similar feedback relationship between miR-128 and *Arpp21* in the adult brain during the suppression of fear-evoked memories (*Lin et al., 2011*). However, mice deficient in *Arpp21* are viable and without a known defect in cortical development (*Rakhilin et al., 2004*; *Davis et al., 2012*).

The disparity we observe between pre-miR-128-2 expression and miR-128 accumulation suggests that a delay in cytoplasmic DICER processing of the precursor contributes to the temporal control of miR-128. For several miRNAs, DICER cleavage is known to be inhibited by precursor-specific RNA binding proteins such as LIN28 in the case of let-7 and miR-9 or DHX36 for miR-134 (*Rybak et al., 2008*; *Bicker et al., 2013*; *Nowak et al., 2014*). A different mechanism, sequestration by the circular RNA sponge CDR1, is thought to control miR-7 (*Hansen et al., 2013*; *Memczak et al., 2013*). The mechanism or mechanisms responsible for post-transcriptional control of miR-128 remain to be determined, however, it appears to help restrict miR-128 accumulation to the cortical plate after neurogenesis and radial migration have occurred. These observations prompted us to test the effects of premature miR-128 expression on radial migration.

Neuronal migration is a complex process necessary for correct cortical lamination and the formation of functional neuronal networks. Previously, three brain-enriched miRNAs (miR-9, miR-132, and miR-137) have been implicated in the regulation of neuronal migration (reviewed in *Evsyukova et al. 2013*). miR-9 and miR-132 cooperate as positive regulators of migration by preventing the expression of the transcription factor FOXP2 (*Clovis et al., 2012*). Similarly, in utero electroporation of miR-137 leads to increased migration of progenitors into the cortical plate due to the ability of miR-137 to stimulate neuronal differentiation (*Sun et al., 2011*). By contrast, we show that miR-128 is a negative regulator of migration and that the onset of miR-128 activity coincides with the termination of upper neuron migration. Manipulating the timing of miR-128 expression interferes with migration and cortical lamination, at least in part through regulation of the transcriptional repressor PHF6.

Like miR-128, the *Phf6* gene is restricted to vertebrates (*Lower et al., 2002*). Based on cross-species comparisons of predicted miR-128 binding sites available at the TargetScan website (*Friedman et al., 2008*), targeting of the *Phf6* mRNA by miR-128 appears to be enhanced in mammals (3 sites), opossum (3 sites), and platypus (2 sites) compared to chicken or frog (no conserved sites). Within the nervous system, mutations in PHF6 have been detected in the developmental disorders Börjeson-Forssmann-Lehmann (BFLS; OMIM 301900) and Coffin–Siris (CSS; OMIM 135900) syndromes (*Lower et al., 2002*; *Tsurusaki et al., 2012*; *Wieczorek et al., 2013*). BFLS is an X-linked recessive intellectual disability disorder associated with epilepsy and other developmental abnormalities. The phenotypic spectrum of CSS phenotypes overlaps BFLS and includes variable intellectual disability and developmental delay. CSS was recently shown to be associated with mutations in several components of SWI/SNF chromatin remodeling complexes in addition to PHF6, strongly suggesting a role for PHF6 in epigenetic regulation (*Santen et al., 2012*; *Tsurusaki et al., 2012*; *Wieczorek et al., 2013*). Furthermore, biochemical evidence has linked PHF6 to several chromatin modifying complexes, including the nucleosome remodeling and deacetylation complex (NuRD) (*Todd and Picketts, 2012*) and the Polymerase associated factor 1 complex (Paf1C) (*Zhang et al., 2013*).

Paf1C has several known functions including histone modification, transcription initiation, and termination (*Jaehning, 2010*).

PHF6 and Paf1C have been implicated in the control of neuronal migration in the mouse. Knockdown of PHF6 during embryonic corticogenesis resulted in impaired upper layer neuron migration characterized by excessive branching of the leading process. Knockdown of PAF1 led to quantitatively similar effects on migration, suggesting that PHF6 acts in the context of Paf1C to facilitate migration (*Zhang et al., 2013*). The reciprocal expression patterns we observe comparing miR-128 and PHF6 during cortical development and neuronal growth in vitro suggest that miR-128 is a significant regulator of PHF6. We also show that the effect of miR-128 on the morphology and final distribution of migrating upper layer progenitors is similar to that reported after PHF6 knockdown (*Zhang et al., 2013*). Moreover, co-expression of PHF6 and miR-128 alleviated this phenotype, indicating that miR-128 is a physiological regulator of PHF6 during corticogenesis. The regulation of SWI/SNF-complex subunit composition by miR-124 provides a precedent for temporal control of epigenetic modifiers by miRNAs during neurogenesis (*Ronan, et al., 2013*). By regulating PHF6, miR-128 may play a similar role for Paf1C or the NuRD complex later in neuronal differentiation. Because premature miR-128 expression inhibited and miR-128 inhibition exaggerated radial migration, the miR-128/PHF6 circuit may play a role in how migrating neurons interpret their position, whether in response to an internal clock, external cues, or cell–cell interactions.

Our results suggest that regulation of PHF6 by miR-128 is important for two interdependent aspects of upper layer neuron maturation in the cortical plate. We show for the first time that miR-128 and PHF6 cooperate in the regulation of dendritic arborization of upper layer neurons. Electrophysiological recordings also show that the balance between miR-128 and PHF6 influences cell autonomous excitability. PHF6 knockdown has previously been shown to increase the excitability of heterotopic neurons that were retained in the white matter due to impaired migration (*Zhang et al., 2013*). Although this finding offers a potential explanation for the cognitive deficits and seizure activity observed in BFLS and CSS, the underlying mechanisms are not yet understood. Comparing the intrinsic properties of neurons expressing either ectopic miR-128 alone or miR-128 together with PHF6, we found that much, but not all, of the difference in intrinsic electrophysiological properties may be directly related to the effects on structural complexity. Layer II/III neurons expressing miR-128 prematurely had reduced complexity of their dendritic arbor, with the most apparent differences observed in their proximal dendrites. This reduction in dendritic complexity was rectified by co-expression of PHF6 and miR-128. Electrophysiological recordings further showed that the input resistance of recorded neurons was increased following miR-128 expression, as would be expected from a reduction in dendritic complexity. Interestingly, premature miR-128 expression also led to a more depolarized resting membrane potential than control cells. This is unlikely to be a direct effect of the morphological changes, and may reflect a reduction in hyperpolarizing leak currents. The net effect of the physiological changes induced by miR-128 was an increase in excitability, reflected by a reduced rheobase and increased firing frequency in response to depolarizing currents. In addition, exogenous PHF6 dampened the effects of miR-128 for all parameters tested. Thus, neuronal excitability is highly sensitive to the precise timing of miR-128 expression and subsequent repression of PHF6 during network formation in vivo. The lack of complete rescue of some parameters by PHF6, however, indicates that additional regulatory targets of miR-128 may contribute to some of the physiological effects we see in post-migratory neurons.

It is interesting to compare our gain-of-function results in cortical neurons to the phenotype observed upon targeted deletion of miR-128 in dopamine responsive neurons of the striatum (D1 neurons) (*Tan et al., 2013*). Loss of miR-128 resulted in heightened excitability that was attributed to the upregulation of ion channels and signal transduction pathways that occurred in the absence of miR-128. In contrast to D1 neurons, there were no significant differences in either the amplitude or the frequency of postsynaptic IPSCs or EPSCs in the cortical neurons we analyzed. Therefore, the regulatory impact of miR-128 may depend on the region and the developmental time point under investigation.

We identify a regulatory interaction between miR-128 and PHF6 that is critical for the proper migration and dendritic outgrowth of upper layer neurons in the developing mouse cortex. These results may have significant relevance for the understanding of cognitive deficits and seizure susceptibility in human patients with mutations in PHF6 and highlight the importance of correct temporal regulation of miR-128 for the establishment of the cortical architecture.

# Materials and methods

## Animals

FMR1 mice were obtained from Charles River (Cologne, Germany), C57Bl/6 mice from the Forschungseinrichtungen für Experimentelle Medizin, Berlin. Animals were handled according to the rules and regulations of the Berlin authorities and the animal welfare committee of the Charité Berlin, Germany.

## Molecular biology reagents and procedures

The expression constructs pre-miR-128-1-RED and pre-miR-128-2-RED contain the respective mouse pre-miRNA sequences together with ≈300 bp upstream and downstream flanking sequences inserted into Intron-RED, the plasmid pEM-157 containing an engineered intron in dsRed (*Makeyev et al., 2007*). The PHF6 sensor construct contains the entire 3'UTR present in NM_032458 cloned downstream of eGFP in a modified peGFP-C1 vector (*Rybak et al., 2008*). The miRNA sensor assay has been described in detail previously (*Rybak et al., 2008*). The PHF6 expression construct contains the PHF6 cDNA cloned into the XhoI and EcoRI restriction sites present upstream of an IRES-GFP cassette in the vector pRS003. PHF6 expression is documented in *Figure 5—figure supplement 3*. Sponge design and cloning strategy are described in *Rybak et al. (2008)*. Sixteen high-affinity binding sites were inserted between the SalI and XhoI ones sites in a modified 3'UTR of peGFP-N1. The repeated sequence is shown in *Supplementary file 1*, as are primer sequences used for all plasmid constructs.

RNA was isolated from dissected forebrain/cortex of the embryonic and post-natal stages and adult brain, from cultured cortical neurons or from transfected HEK-293 cells (Lipofectamine 2000) using *TRIzol* (Life Technologies, Carlsbad, CA) according to manufacturer's instruction. For qRT-PCR of mRNA, cDNA was synthetized using RevertAid Premium Reverse Transcriptase (Thermo Scientific, Valencia, CA) followed by amplification using RT2 SYBR Green (Sabio Sciences/Qiagen, Venlo, Netherlands) according to manufacturer's instructions. GAPDH was used for normalization of primary cortical neuron samples and Oaz1 for brain samples. Quantification of miRNA expression made use of miRNA TaqMan Assays for miR-128 normalized against sno135 (Probe Set ID:000589 and ID:1230, Life Technologies).

Western blotting followed standard procedures using HeLa, HEK-293 or primary cortical lysates prepared in 1% NP-40, 20 mM Hepes pH 7.9, 350 mM NaCl, 1 mM MgCl$_2$, 0.5 mM EDTA, 0.5 mM EGTA, 50 mM NaFl, 1 mM DTT with the addition of protease inhibitor cocktail set I (Calbiochem/EMD Millipore, Schwalbach, Germany). An ImageQuant LSA 4000mini (GE Healthcare, Little Chalfont, United Kingdom) was used for detection, quantification by normalization to loading controls was done using Fiji software.

## Northern blots

Electrophoresis and blotting are described in *Rybak et al. (2009)*; *Smirnova et al. (2005)*. For hybridization 20 μM LNA probe (Exiqon A/S, Vedbaek, Denmark) was radioactively labeled using 60 μCi [gamma$^{32}$-P] ATP and T4 Polynucleotide Kinase (Fermentas/Thermo Scientific). The labeled probes were diluted in 5 ml hybridization buffer (250 mM Na$_2$HPO$_4$ (pH 7.2), 7% SDS, 1 mM EDTA, 1% BSA). The membrane was incubated in a rotating hybridization oven at 46°C and then washed twice in 2× SSPE, 0.1% SDS and twice in 0.5× SSPE, 0.1% SDS. The signal was detected by autoradiography.

## In situ hybridization

In situ hybridization was performed using 5' and 3' digoxygenin labeled LNA probes (Exiqon A/S) essentially as described in *Silahtaroglu et al. (2007)*. Embryonic and early postnatal brain tissue was collected at the appropriate stage and fixed overnight in 4% PFA, adult brain tissue was collected after perfusion. The tissue was hybridized with double digoxigenin labeled LNA probes (Exiqon A/S) at the suggested hybridization temperature. Anti-digoxigen antibodies and any primary antibodies to detect proteins of interest were incubated simultaneously overnight at 4°C. Protein detection was performed first with appropriate labeled secondary antibodies followed by the enzymatic reaction to detect the miRNA. NBT/BCIP (Roche tablets) or Fast red (Roche tablets) were used, according to manufacturer's instructions (Hoffmann-La Roche, Basel, Switzerlad), to detect miRNAs for bright field or fluorescence microscopy, respectively.

For Phf6 mRNA detection the tissue was hybridized at the suggested temperature using a custom LNA probe (Exiqon, see Table 1) with 5' biotin and 3' biotin-TEG labels. Anti-streptavidin-HRP antibody

(1:500) was incubated overnight at 4°C. Then the Tyramide Signal Amplification (TSA)-Cyanine 3 system (Perkin Elmer, Waltham, MA) was used according to manufacturer's instructions: the fluorophore was diluted 1:50 in Amplification buffer and developed in the dark for 7 min.

### Nissl staining

Cryosections were incubated in potassium sulfide solution (50% Potassium disulfide dissolved in water) for 15 min, washed twice in water and incubated in cresyl violet solution (1.5% cresyl violet dissolved in acetate buffer) for 30 min. Slices were washed for 1 min in Acetate buffer (0.01 M Sodium acetate, 0.01 M Acetic acid), 30 s in Differentiation buffer (500 ml water, 700 µl Acetic acid), and rinsed once in water. The slides were dehydrated and mounted.

### Fluorescent intensity measurement

After in situ hybridization, using the NBT/BCIP detection method, the sections were imaged using an Olympus BX51 microscope and 40× objective. The colors of the bright field image were inverted in Fiji and the resulting image was used to measure the fluorescent intensity. The area of interest was contoured using the Polygon selection tool. The integrated density, mean fluorescence, and the area were measured. In the same image, an unstained region was contoured and measured for background substraction. The corrected total cell fluorescence (CTCF) was calculated using the formula:

CTCF = Integrated density − (area of selected region × mean of background). The fluorescence of IZ and VZ/SVZ were normalized to the fluorescence of the CP in each image. The normalized values were used for the analysis. At least three slices per brain and three brains per condition were analyzed. The statistical test used was One-Way ANOVA.

### PHF6 antibody staining

Embryonic brain tissue was collected at the appropriate stage and fixed in 2% PFA for 6 hr. Cryosections were not post-fixed but directly incubated in blocking buffer (1× PBS, 0.25% Triton X, 0.1% Tween 20, 3% BSA). The sections were incubated overnight at 4°C with anti-PHF6 antibody (BETHYL A301-451A 1:100, Bethyl Laboratories, Montgomery, TX). Antibody specificity is documented in *Figure 5—figure supplement 3A*. For the detection, the tissue was incubated for 1 hr at room temperature with anti-rabbit secondary antibody-HRP conjugate followed by TSA Cyanine 3 system detection according to manufacturer's instructions (Perkin Elmer). The fluorophore was diluted 1:50 in Amplification buffer and developed in the dark for 7 min.

### In utero electroporation

In utero electroporation of NMRI mice was performed as described in *Saito (2006)* with minor modifications. A 300 ng/µl solution of pre-miR-128-1-RED, pre-miR-128-2-RED or control Intron-RED plasmids and/or IRES-GFP control or PHF6-GFP vector at 150 ng/µl was injected in one lateral ventricle. E15.5 embryos were electroporated using 6 pulses of current at 35 mV. The resulting embryos or pups were processed for immunohistochemistry (migration analysis, marker detection) or electrophysiology.

### Migration analysis

The migration analysis was assessed at P7 on 50-µm brain slices. The slices were collected from the beginning of the corpus callosum to the middle of the hippocampus. Floating slices were stained for detection of dsRed (Abcam ab62341 at 1:150, Cambridge, United Kingdom) and GFP (Abcam ab13970 at 1:500). The primary antibodies were dissolved in blocking solution (1× PBS, 0.25% Triton-X, 0.1% Tween-20, 3% BSA). The slices were incubated in primary antibody overnight at room temperature with shaking. Secondary antibodies were incubated 2 hr at room temperature on a shaker. The mounted sections were imaged using a Leica SL confocal microscope with a 10× objective. A grid consisting of 10 bins was applied to the images, positioning the beginning of the first bin at the beginning of layer II and the end of the tenth bin at the end of layer VI, as determined by visual analysis of nuclear staining (DRAQ5), essentially as described (*Rosário et al., 2012*). When necessary more than one adjacent grid was applied to cover the entire electroporated region. Neurons within each bin were counted using the Cell counter plugin for Fiji. An average of five sections from at least three independent brains per condition was analyzed. The number of neurons in each bin was normalized first for individual brains and then the normalized value was used as n = 1 per condition. The data were analyzed in Prism 5.0 using Two-way ANOVA.

## Layer marker detection and counting

Intron-RED and pre-miR-128-2-RED electroporated pups from the same litter were analyzed at P0 (born E20). 10 μm cryosections were stained for dsRed (Abcam ab62341 1:150), Cux1 (Santa Cruz Biotechnology, sc-13024 1:150, Heidelberg, Germany), and Ctip2 (Abcam ab18465 1:500). The slices were imaged using a Leica SL confocal microscope with 40× objective. Using the Cell Counter plugin for Fiji both the total number of electroporated neurons in the cortical plate and the number of electroporated neurons positive for either Cux1 or Ctip2 was counted. The number of neurons positive for the layer marker was normalized to the total number of electroporated neurons. Three independent brains electroporated with pre-miR-128-2-RED and one brain electroporated with Intron-RED were analyzed and for each layer marker at least three slices per brain were counted.

## P0 migration morphology

The analyzed P0 brains (born E19) from Intron-RED + pRS003 (n = 3) and pre-miR-128-2-RED (n = 4) electroporated animals were from the same litter. 60-μm sections were stained for dsRed (Abcam ab62341 1:150) and GFP (Abcam ab13970 1:500). Nuclear staining was obtained with DRAQ5 (Biostatus, Shepshed, United Kingdom). Images were taken using a Leica SL confocal microscope. The overview was taken as a single image with 10× objective. Images for reconstruction of migrating neurons were taken with a 40× objective and a 1 μm step Z-stack. The deep layers were defined using nuclear stain and a pool of migrating neurons within the deep layers was reconstructed using the Fiji plugin Simple Neurite tracer. The number of branches and filopodia (excluding the trailing process) was counted. To distinguish between branch and filopodium a cut-off of 5 μm was used.

## Electrophysiological recording

Acute brain slices were prepared from P15 mice after electroporation as described in the text. Slice preparation, recordings, visualization of the neurons, and data analysis were performed as described previously (*Booker et al., 2013*). In brief, 300-μm thick coronal slices including the somatosensory cortex were prepared in ice-cold carbogenated sucrose-substituted artificial cerebrospinal fluid (ACSF; in mM: 87 NaCl, 2.5 KCl, 25 NaHCO$_3$, 1.25 NaH$_2$PO$_4$, 25 glucose, 75 sucrose, 7 MgCl$_2$, 0.5 CaCl$_2$, 1 Na-Pyruvate, 1 Ascorbic Acid), left to recover at 35°C for 30 min, then stored at room temperature.

Whole-cell patch clamp recordings were performed in a submerged recording chamber superfused with carbogenated recording ACSF (in mM: 125 NaCl, 2.5 KCl, 25 NaHCO$_3$, 1.25 NaH$_2$PO$_4$, 25 glucose, 1 MgCl$_2$, 2 CaCl$_2$, 1 Na-Pyruvate, 1 Ascorbic Acid) at 32–34°C, from visually identified GFP-positive neurons within the electroporated region of the somatosensory cortex, using a Multiclamp 700B amplifier (Molecular Devices, Sunnyvale, CA). Patch pipettes were filled with a K-gluconate based intracellular solution (in mM: 130 K-Gluc, 10 KCl, 2 MgCl$_2$, 10 EGTA, 10 HEPES, 2 Na$_2$-ATP, 0.3 Na$_2$-GTP, 1 Na$_2$-Creatinine and 0.1% biotinylated-lysine (Biocytin, Invitrogen/Life Technologies), pH 7.3, 290–310 mOsm), resulting in a pipette resistance of 2–5 MΩ. Voltage signals were digitized at 10 kHz (NI-DAQ, National Instruments, Newbury, UK), acquired with WinWCP software (J Dempster, Strathclyde University) and analysed offline using Stimfit (C Schmidt-Hieber; www.stimfit.org).

The intrinsic physiology of neurons was characterized in current-clamp mode, with a family of hyperpolarizing to depolarizing current pulses (−250 to 250 pA, 50 pA steps, 500 ms duration); determining the current–voltage relationship, I$_h$ mediated voltage sag and action potential (AP) discharge frequency. Small hyperpolarizing current pulses (−10 pA, 500 ms duration) were applied to assess the input resistance (RI) of the recorded neurons. Membrane potential (VM) was calculated as the 50 ms baseline prior to the small hyperpolarizing step.

Following intrinsic characterization, outside-out patches were formed and biocytin was allowed to fill the cell for an additional 15 min. Slices were immersion fixed in 4% formaldehyde in 0.1 M phosphate buffer (PB) overnight at 4°C. Slices were copiously rinsed in PB and the filled cells visualized with Avidin-conjugated Alexa-Fluor-647 (Invitrogen/Life Technologies; 1:1000), in PB containing 0.3% Triton X-100 and 0.05% NaN$_3$, overnight at 4°C. Slices were subsequently rinsed in PB and mounted on glass slides, with a 300 μm agar spacer to prevent compression of the slices after cover-slipping. The slices were imaged using Leica SL confocal (1024 × 1024 resolution) using ×20 objective and 200 Hz speed. The step between stacks was 1 μm.

## Neuronal reconstruction and morphometric analysis

For analysis at P0 60 μm slices were prepared from single litters of electroporated animals and processed for immunostaining with dsRed and eGFP antibodies plus DRAQ5 nuclear stain. Images were taken using a Leica SL confocal microscope, for reconstruction a 40× objective and Z-stack step of 1 μm was used. Nuclear staining was used to identify the deep layers of the cortical plate, individual neurons were reconstructed with the Fiji plugin Simple Neurite tracer. Quantification was essentially as described in *Guerrier et al. (2009)*. For analysis at P21 electroporated animals were sacrificed, perfused, and 100-μm slices were prepared. Electroporated neurons were visualized by staining with dsRed antibody. Z-stack images were taken with an inverted epifluorescence microscope (Olympus IX81) with a 1 μm stack and reconstructed as above. For P15 neurons, after recording outside-out patches were formed and cells were filled with biocytin for 15 min. After overnight fixation by immersion in 4% formaldehyde in 0.1 M phosphate buffer (PB) at 4°C, the filled cells visualized with Avidin-conjugated Alexa-Fluor-647 (Invitrogen/Life Technologies; 1:1000), in PB containing 0.3% Triton X-100 and 0.05% $NaN_3$, overnight at 4°C. After rinsing in PB and mounting on glass slides with a 300 μm agar spacer the slices were imaged using a 20× objective and a Leica SL confocal microscope at 200 Hz. The step between stacks was 1 μm at a resolution of 1024 × 1024. Neurons were reconstructed using the Simple Neurite Tracer plugin. Sholl analysis was performed on 3-D reconstructions using the Sholl analysis plugin for Fiji. The radius of the first concentric sphere was set at 7.5 μm and the increase between radii was 5 μm. The data set for Sholl analysis at P15 and P21 are provided in *Supplementary file 2* and *Supplementary file 3*, respectively.

## Statistical analysis

Statistical analysis was performed using Prism 5.0, when indicated multi-group comparisons were analyzed by Two-way ANOVA with the Bonferroni posttest; when comparing two groups a Student's unpaired *t*-test was employed as indicated in each legend. Significance is denoted in the Figures as: ***$p < 0.001$; **$p < 0.01$; *$p < 0.05$; ns, not significant.

## Acknowledgements

We would like to thank all members of the Tarabykin and Wulczyn laboratories for scientific discussions, and Daniel Richter for expert technical assistance. We would like to thank Franck Polleux, Julien Courchet, Theresa Köbe, Steffen Schuster, and Marta Rosário for advice on the quantification of neuronal morphology.

## Additional information

### Funding

| Funder | Grant reference number | Author |
|---|---|---|
| Deutsche Forschungsgemeinschaft | Graduate School 1123 | Eleonora Franzoni |
| Deutsche Forschungsgemeinschaft | Collaborative Research Program 665 | Eleonora Franzoni, Frederick Rehfeld, Heiko R Fuchs, F Gregory Wulczyn |
| Deutsche Forschungsgemeinschaft | Excellence Initiative 257 | Imre Vida |

The funders had no role in study design, data collection and interpretation, or the decision to submit the work for publication.

### Author contributions

EF, Conception and design, Acquisition of data, Analysis and interpretation of data, Drafting or revising the article; SAB, Acquisition of data, Analysis and interpretation of data, Drafting or revising the article; SP, FR, SG, SS, HRF, Acquisition of data, Analysis and interpretation of data; VT, Conception and design, Drafting or revising the article; IV, FGW, Conception and design, Analysis and interpretation of data, Drafting or revising the article

### Ethics

Animal experimentation: All experiments were conducted according to the European and German laws, in conformance with the Animal Welfare Act and the European legislative Directives

86/609/EEC 2010/63/EU from 2010 on as updated in 2013. The animal welfare committee of the Charité, Berlin, approved and supervised the experiments performed under the experimental license number T01012/11. All surgery was performed using Isofluran as anesthetic and Temgesic for analgesia as required to minimize suffering.

## Additional files

### Supplementary files

• Supplementary file 1. Contains Tables of LNA probe sequences, primers used in reporter, and expression plasmid cloning as well as qRT-PCR analysis.

• Supplementary file 2. Related to *Figure 7*: data set used for Sholl analysis at P15.

• Supplementary file 3. Related to *Figure 7—figure supplement 2*: data set used for Sholl analysis at P21.

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
