## [Decision Letter]

Thank you for sending your work entitled “miR-128 regulates neuronal migration, outgrowth and intrinsic excitability via the intellectual disability gene *Phf6*” for consideration at *eLife*. Your article has been favorably evaluated by a Senior editor, a Reviewing editor, and 2 reviewers.

The Reviewing editor and the reviewers discussed their comments before we reached this decision, and the Reviewing editor has assembled the following comments to help you prepare a revised submission.

The reviewers and the Reviewing editor agreed that your results are significant and interesting. The identification of a new function for miR128 in neuronal migration, dendritic branching and neuronal excitability as well as the characterization of *Phf6* as a potential target of miR128 in this context provide new insights into the molecular mechanisms underlying cortical circuit development. However, the reviewers also raised a number of major weaknesses in the paper that need to be addressed with new experiments in order to improve the main conclusions of the paper.

Major comments:

1) The study does not provide compelling evidence that *Phf6* is a target of miR-128 in the developing cortex. miR-128 is shown to suppress *Phf6* expression in HEK-293 cells, and *Phf6* rescues miR-128 ectopic/overexpression phenotypes in cortical neurons. However, whether miR-128 normally regulates *Phf6* expression in the developing cortex is not addressed (and whether and how *Phf6* is expressed is not examined). Therefore the functional study of a putative miR-128-*Phf6* pathway in cortical development does not stand on solid ground.

2) The functional analysis of miR-128 is only based on gain of function/overexpression experiments. The authors need to provide loss-of-function evidence downregulating mir-128 to demonstrate that it is required for the migration and dendritic branching phenotypes described. The phenotypes resulting from electroporation of miR-128 are analysed at two stages: i) in cortical plate neurons where mature miR-128 is normally expressed, and where exogenous miR-128 is therefore overexpressed; ii) in migrating IZ neurons where mature miR-128 is normally not expressed and where exogenous miR-128 is therefore expressed ectopically. This is assuming that electroporation of pre-miR-128-2 results in production of mature miR128, which seems logical since it has a biological effect, although this is not shown. There is also no explanation of why exogenous pre-miR-128-2 has an activity in migrating neurons and is therefore presumably processed into the mature form, while endogenous pre-miR-128-2, which is also expressed in these cells, is not.

3) The authors’ arguments regarding the layer specificity of miR128-2 expression in the developing neocortex, especially in the absence of a careful quantitation of the levels of fluorescent signal is far from convincing. The authors should measure miR124 and miR128-2 expression levels in the VZ, SVZ and IZ and normalized their expression levels to those of the cortical plate, in order to test if miR124 and miR128-2 would still be significantly different in VS, SVZ and IZ.

---

## [Author Response]

For the sake of argument I will address the main points in reverse order.

*3) The authors’ arguments regarding the layer specificity of miR128-2 expression in the developing neocortex, especially in the absence of a careful quantitation of the levels of fluorescent signal is far from convincing. The authors should measure miR124 and miR128-2 expression levels in the VZ, SVZ and IZ and normalized their expression levels to those of the cortical plate, in order to test if miR124 and miR128-2 would still be significantly different in VS, SVZ and IZ*.

To meet this request we now compare colorimetric staining for pre-miR-128-2, miR-128 and miR-124 at E14.5 and E16.5 in greater detail (representative images are shown in Figure 2). Using this expanded dataset we have performed the quantifications as requested and the results fully support our original interpretation.

In our view, the critical point for our manuscript is the discrepancy between pre-miR-128-2 and miR-128 expression patterns in the VZ, SVZ and IZ documented in the original Figure 2 and S3. These images are now presented in Figure 2—figure supplement 2, where they can be directly compared to miR-124. Quantification of the new dataset at E14.5 shows that average pre-miR-128-2 staining intensity is *higher* in the VZ/SVZ and in the IZ relative to the CP (approximately 3:1 and 1.5:1, respectively). The opposite is true for miR-128, staining intensity relative to the CP is *reduced* in the VZ/SVZ and IZ, respectively (Figure 2). A similar pattern is seen at E16.5: staining intensity for pre-miR-128-2 in the VZ/SVZ and IZ is slightly reduced compared to the CP (approximately 08:1). In contrast, for miR-128 staining intensity in the VZ/SVZ and IZ strongly reduced compared to the CP (approximately 01:1, Figure 2). We have also replaced the original images comparing pre-miR-128-2 and miR-128 at E18.5 (Figure 3—figure supplement 1) with improved sections to demonstrate that the disparity in precursor and mature miRNA staining is maintained at the end of embryonic cortical neurogenesis.

In addition, we consistently observe miR-124 positive cells in the IZ during corticogenesis. In our view this is interesting in its own right but is not a condition for our hypothesis to be correct. Therefore we did not document this effect in detail in the original submission. At the reviewers’ request we now include miR-124 in the new analysis (Figure 2). The bright field images show, in contrast to miR-128, individual miR-124 positive cells scattered in the IZ at E14.5 and E16.5. Quantification of these images confirm that the relative staining intensity for miR-124 in the IZ is higher than the VZ/SVZ at E14.5 and E16.5 (Figure 2), consistent with the original data. This is in clear contrast to miR-128, for which staining intensity in the subcortical zones is more uniform. These results suggest that the mechanism involved in delaying miR-128 biogenesis is not shared by miR-124, the paradigm miRNA in the regulation of neurogenesis.

Within the cortical plate itself, it is our impression that upper layer neurons express miR-128 at lower relative levels than either pre-miR-128 or miR-124. This visual impression is reflected in the quantitative analysis but is not statistically significant.

We hope this new dataset and the improved presentation satisfies the reviewers’ concerns.

*2) The functional analysis of miR-128 is only based on gain of function/overexpression experiments. The authors need to provide loss-of-function evidence downregulating miR-128 to demonstrate that it is required for the migration and dendritic branching phenotypes described*.

In my view it is not necessarily self-evident that inhibiting miR-128 during migration or outgrowth, as requested, should be expected to yield an easily predicted phenotype. Introducing miR-128 prematurely into miR-128^-^ progenitors, as we do in our experiments is analogous to a knockdown experiment, in our case of miR-128 targets that are normally inhibited at a later stage but serve a function in migrating neurons. The reverse experiments may not be informative (in the case of a knockdown overexpressing the gene of interest, in our case inhibiting the miRNA).

That said, we have performed the suggested experiments and can comply with the reviewers’ request by showing that “sponge”-mediated inhibition of miR-128 leads to a significant overmigration of upper layer neurons in vivo under the same experimental conditions we used to show that premature miR-128 impairs migration. This new data is included in Figure 3. We believe this “mirror image” phenotype substantially satisfies the reviewers’ request and should lead to interesting mechanistic insights in the future.

We also performed Sholl analysis of “sponge”-knockdown cells at P15. As the reviewers can see in Figure 8, the small increase in dendritic complexity we observe is suggestive but not statistically significant. There are several possible explanations for this lack of phenotype. First, postnatal expression of miR-128 during outgrowth is very high compared to the situation in migrating cells at the onset of miR-128 expression (Figure 1 and Figure 2). Therefore migration may be more sensitive to the sponge than outgrowth, due to the high but limited number of sponge binding sites expressed in each cell (see Broderick and Zamore, 2014 for a discussion of the limitations of competition strategies). Alternatively, since the outgrowth phenotype was substantially mediated by *Phf6* (as seen in the rescue experiments), it is possible that inappropriate *Phf6* overexpression due to the sponge does not cause an obvious morphological defect.Author response image 1.Reconstructed neurons obtained by biocytin filling of cells electroporated with an eGFP control (dark green), or an anti-miR-128 sponge (light green) construct are shown in A and B. The results of Sholl analysis are presented in C, comparing eGFP control (dark green) and anti-miR-128 sponge (light green). Error bars represent SEM.

All in all, we believe the new results obtained in the migration assay greatly strengthen the case for miR-128 in the control of migration. We hope we have satisfied the reviewers’ concerns.

*The phenotypes resulting from electroporation of miR-128 are analysed at two stages: i) in cortical plate neurons where mature miR-128 is normally expressed, and where exogenous miR-128 is therefore overexpressed; ii) in migrating IZ neurons where mature miR-128 is normally not expressed and where exogenous miR-128 is therefore expressed ectopically. This is assuming that electroporation of pre-miR-128-2 results in production of mature miR128, which seems logical since it has a biological effect, although this is not shown. There is also no explanation of why exogenous pre-miR-128-2 has an activity in migrating neurons and is therefore presumably processed into the mature form, while endogenous pre-miR-128-2, which is also expressed in these cells, is not*.

We point out that we do not know whether the outgrowth phenotype is triggered by premature miR-128 before or after the neurons reach the cortical plate. Either way, the reviewers are correct that we are tacitly assuming that the inhibitory mechanism that blocks processing of endogenous pre-miR-128-2 is titrated out/overloaded when we introduce miR-128 expression constructs into progenitors. We believe this assumption is reasonable: as a precedent, it is well known that ectopic expression of the let-7 precursor in pluripotent cells results in biogenesis of mature let-7 despite the presence of the inhibitory Lin28 protein (see for example [34]). However, we now provide direct experimental support: we have performed in situ hybridization for miR-128 in the cone of pre-miR-128-2-RED electroporations and can clearly show that dsRed-positive cells in the intermediate zone stand out as miR-128-positive among the otherwise negative cells of the IZ. Similarly positioned cells in control electroporations performed in parallel do not express miR-128. This data is provided in Figure 7—figure supplement 2. We agree that this demonstration should significantly strengthen the paper and thank the reviewers for pointing out this omission.

*1) The study does not provide compelling evidence that Phf6 is a target of miR-128 in the developing cortex. miR-128 is shown to suppress Phf6 expression in HEK-293 cells, and Phf6 rescues miR-128 ectopic/overexpression phenotypes in cortical neurons. However, whether miR-128 normally regulates Phf6 expression in the developing cortex is not addressed (and whether and how Phf6 is expressed is not examined). Therefore the functional study of a putative miR-128-Phf6 pathway in cortical development does not stand on solid ground*.

We have addressed this legitimate concern in two ways. First, we were somewhat remiss in citing published expression studies on *Phf6* that have documented *Phf6* expression in the ventricular, subventricular and intermediate zones in the relevant periods of corticogenesis (52; 49). Indeed, the previously published knockdown experiments showing that in utero electroporation of shRNAs directed against *Phf6* impairs the migration of cortical neurons would not make sense if this were not the case (52). To further document *Phf6* expression patterns, we are providing new in situ data for *Phf6* using both a highly specific LNA-based probe for mRNA detection and a newly available antibody (now included in Figure 5). The specificity of the antibody is confirmed by western blot (Figure 5—figure supplement 3).

The results at the mRNA and protein level confirm the previously published results, and show prominent staining for *Phf6* in the VZ/SVZ and in the IZ in the relevant period of corticogenesis (E14.5 and E16.5). These experiments allow a direct comparison to miR-128 at the relevant time points. The pattern of miR-128 expression is reciprocal (differential expression of miR-128 in each cortical zone is described in more detail and quantified in the new version of Figure 2). This new in situ data is consistent with, and supported by, the temporal profile of *Phf6* mRNA (Figure 5, qRT-PCR) and mature miR-128 (Figure 1, Northern blot) in brain development and during neuronal maturation in culture (Figure 5, qRT-PCR). The expression patterns documented in the revised version of Figure 5 are therefore in complete accord with our hypothesis and with previous work on the role of *Phf6* in corticogenesis (52): *Phf6* is expressed in the relevant layers during generation and migration of upper layer neurons. miR-128 is excluded from the VZ/SVZ and IZ, with accumulation restricted to the cortical plate (Figure 2). We believe we have convincingly shown that the temporal dynamics of *Phf6* and miR-128 regulation match their proposed roles in neuronal migration and outgrowth. The results also document the presence of *Phf6* in the cell populations we target in our ectopic miR-128 electroporations, supporting our view that premature miR-128 expression could potentially interfere with the known requirement for *Phf6* during migration as reported by Zhang et al. (52).

The second part of the comment questions if we have done enough to show that *Phf6* is indeed a regulatory target of miR-128. We point out that a direct demonstration of *Phf6* regulation by miR-128 in the developing cortex is technically quite difficult. Generally speaking, in the highly analogous case of shRNA knockdowns it is not usually required to show that the shRNA construct reduces the target protein expression in vivo after in utero electroporation. It should suffice to demonstrate the efficacy and specificity of the shRNA in cell culture and that the biological effect is rescued in vivo by an shRNA-insensitive expression construct. What we have shown is analogous: we show that a *Phf6* construct lacking the miR-128 recognition sequences in the 3’UTR rescues the migration, outgrowth and electrophysiological phenotypes we see with ectopic miR-128. We show that the three conserved miR-128 binding sites in the *Phf6* 3’UTR (now described in Figure 5—figure supplement 1) are recognized in a reporter assay (now shown in Figure 5). This is not unexpected, we point out that *Phf6* is the sixth highest ranked predicted target for miR-128 in the PicTar database PicTar (19). We also show that introduction of miR-128 into the miR-128-negative HEK-293 cell line reduces the level of endogenous *Phf6* at the protein level (Figure 5). We now show the results of the same experiment in HeLa cells, in which ectopic miR-128 but not non-targeting control miRNAs clearly reduce *Phf6* protein levels (Figure 5).

For the benefit of the reviewers, we would like to expand on this data at some length, presenting unpublished work from our lab.

The first indication that *Phf6* is a miR-128 target came from the miRNA prediction algorithms TargetScan, PicTar and DianaMicroT. All three identify three potential binding sites that are present in the 3’UTR of human and mouse *Phf6* mRNAs (see Figure 5—figure supplement 2). DianaMicroT recognizes one additional, lower affinity binding site. In TargetScan two of the sites are designated “poorly conserved”. This is because, interestingly, they are not present in chicken or Xenopus mRNAs. However, the three sites are conserved in all mammalian mRNAs.

These observations strongly suggest that *Phf6* is an evolutionarily conserved target in mammals for miR-128, with the presence of multiple binding sites further suggesting a relatively strong regulatory effect.

We have previously performed an unbiased screen for miR-128 response genes in the P19 embryocarcinoma cell line (our unpublished data). 160 transcripts were found to be downregulated (cutoff 0.5-fold). As an indication of the specificity of targeting, analysis of this list for miRNA target enrichment (Web-Gestalt) reveals that targets for miR-128 and miR-27 are the only miRNAs showing significant enrichment, as expected (P = 2.3e-08 or 3.2e-08, Bonferonni correction, respectively). The seed regions of miR-128 and miR-27 are highly similar and the two miRNAs are expected to target overlapping sets of mRNAs. This indicates that at least some of the mRNAs identified in this experiment are likely to be direct targets. In addition to *Phf6*, two of the best characterized miR-128 target mRNAs in the nervous system, *Casc3/Mln51* (4) and *Szrd1* (45) were also detected in this screen. In the microarray experiment *Phf6* was downregulated 0.441-fold, very similar to the degree of inhibition we detect at the mRNA and protein level in HEK-293 cells as shown in Figure 5, compared to 0.284-fold for *Szrd1* with 4 predicted binding sites and 0.478-fold for *Casc3* with two predicted sites. These results have been confirmed by RT-PCR and in the case of *Szrd1* a reporter assay using the cloned 3’UTR (our unpublished observations).

Finally, after submission of our manuscript, a study was published providing evidence for targeting of *Phf6* by miR-128 in T cells, one of the few tissues outside the nervous system displaying high levels of miR-128 expression (Mets et al., 2014).

Taken together, we believe the evidence for regulation of *Phf6* by miR-128 is strong. However, we have made every effort to directly prove that *Phf6* is downregulated in vivo under our experimental conditions. For this purpose we employed FACS to isolate control Intron-RED and pre-miR-128-2-RED cells at E18.5 shortly after electroporation at E15.5 (Figure 9). Unfortunately, yields of cells were low (≈200,000/litter), leading to experimental variability in the downstream processing required to produce cDNA for qRT-PCR analysis. We observe a trend of reduced expression of *Phf6* in miR-128-2 electroporated cells compared to control. However, the difference in 4 independent samples derived from cells pooled from independent electroporations did not reach statistical significance. For comparison, we chose the mRNA for *Szrd1/C1orf144* as positive control, which as discussed above is an experimentally confirmed miR-128 target (45) and the top-ranked predicted mRNA target for miR-128 (Targetscan, PicTar). The trend we observed was comparable to *Phf6* but also not statistically significant.Author response image 2.qRT-PCR performed on FAC-sorted cells obtained after electroporation at E15.5 with an Intron-RED control plasmid (black bars) or the miR-128-2 expression plasmid pre-miR-128-1-RED (gray bars). dsRed positive cells were collected, RNA isolated and qRT-PCR performed using *Oaz1* as reference mRNA. *PHf6*, Szrd1 were readily detected, quantification is shown. *Szrd1* as a known target of miR-128 is the positive control, *β*-actin the negative control. Although a clear trend is observed, the reduction of *PHf6* mRNA in response to ectopic miR-128 is not statistically significant (unpaired t-test).

We point out that, unlike siRNAs, for miRNAs qRT-PCR is likely to underestimate the degree of repression at the protein level. We believe that the technical difficulty of the experiment and the attendant variability makes it difficult to achieve significance given the relatively modest effect we are trying to capture: under ideal conditions an approximately 50% reduction in the reporter assay (Figure 5) and in protein levels in HEK-293 cells (Figure 5). We hope the reviewers agree that this experiment is susceptible to a Type II error. We would also like to say that we consider FACS analysis a fairly heroic effort and hope that it does not become the gold standard, as it requires considerable animal experimentation and the availability of an excellent FACS facility.

We have considered alternative approaches: we see no way to perform quantitative western blots on the limited number of cells obtained by FACS. Based on our experience with other miRNAs, we do not believe staining of miR-128 positive or deficient cells for *Phf6* would be quantitatively rigorous, particularly since all available *Phf6* antibodies require amplification using the TSA system.

In summary, *Phf6* is a predicted and validated regulatory target for miR-128 as shown in multiple assays. The largely reciprocal expression patterns of *Phf6* and miR-128 are consistent with a role for miR-128 in the regulation of *Phf6* during radial migration of cortical neurons. We show for the first time that excess miR-128 during development perturbs migration, outgrowth and intrinsic activity of upper layer cortical neurons. We show that exogenous *Phf6* can rescue these phenotypes. These results are consistent with previous reports on *Phf6* activity in migration and extend the influence of *Phf6* to outgrowth and concomitant measures of activity. We hope that the evidence for this regulatory interaction is now on much more solid ground.